# Dual Inhibitory Potential of Conessine Against HIV and SARS-CoV-2: Structure-Guided Molecular Docking Analysis of Critical Viral Targets

**DOI:** 10.3390/v17111435

**Published:** 2025-10-29

**Authors:** Ali Hazim Abdulkareem, Meena Thaar Alani, Sameer Ahmed Awad, Safaa Abed Latef Al-Meani, Mohammed Mukhles Ahmed, Elham Hazeim Abdulkareem, Zaid Mustafa Khaleel

**Affiliations:** 1Department of Biotechnology, College of Science, University of Anbar, Ramadi 31001, Iraq; ali.hazim@uoanbar.edu.iq (A.H.A.); sc.safaa-meani@uoanbar.edu.iq (S.A.L.A.-M.); zaidmustafa683@gmail.com (Z.M.K.); 2College of Dentistry, University of Anbar, Ramadi 31001, Iraq; meena.thayir@uoanbar.edu.iq; 3Department of Medical Laboratories Techniques, College of Health and Medical Technology, University of Al Maarif, Al Anbar 31001, Iraq; sameer.msc1981@gmail.com; 4Department of Oral and Maxillofacial Surgery, College of Dentistry, University of Anbar, Ramadi 31001, Iraq; den.elham.h@uoanbar.edu.iq

**Keywords:** conessine, HIV-1, SARS-CoV-2, molecular docking, spike protein RBD, HIV protease, molecular dynamics, ADMET profiling, steroidal alkaloid, broad-spectrum antiviral

## Abstract

Human immunodeficiency virus (HIV-1) and SARS-CoV-2 continue to co-burden global health, motivating discovery of broad-spectrum small molecules. Conessine, a steroidal alkaloid, has reported membrane-active and antimicrobial properties but remains underexplored as a dual antiviral chemotype. To interrogate conessine’s multi-target antiviral potential against key enzymatic and entry determinants of HIV-1 and SARS-CoV-2 and to benchmark performance versus approved comparators. Eight targets were modeled: HIV-1 reverse transcriptase (RT, 3V81), protease (PR, 1HVR), integrase (IN, 3LPT), gp120–gp41 trimer (4NCO); and SARS-CoV-2 main protease (M^pro^, 6LU7), papain-like protease (PL^pro^, 6W9C), RNA-dependent RNA polymerase (RdRp, 7BV2), spike RBD (6M0J). Ligands (conessine; positive controls: dolutegravir for HIV-1, nirmatrelvir for SARS-CoV-2) were prepared with standard protonation, minimized, and docked using AutoDock Vina v 1.2.0exhaustiveness 4; 20 poses). Binding modes were profiled in 2D/3D. Protocol robustness was verified by re-docking co-crystallized ligands (RMSD ≤ 2.0 Å). Atomistic MD (explicit TIP3P, OPLS4, 300 K/1 atm, NPT; 50–100 ns) assessed pose stability (RMSD/RMSF), pocket compaction (Rg, volume), and interaction persistence; MM/GBSA provided qualitative energy decomposition. ADMET was predicted in silico. Conessine showed coherent, hydrophobically anchored binding across both viral panels. Best docking scores (kcal·mol^−1^) were: HIV-1—PR −6.910, RT −6.672, IN −5.733; SARS-CoV-2—spike RBD −7.025, M^pro^ −5.745, RdRp −5.737, PL^pro^ −5.024. Interaction maps were dominated by alkyl/π-alkyl packing to catalytic corridors (e.g., PR Ile50/Val82, RT Tyr181/Val106; M^pro^ His41/Met49; RBD L455/F486/Y489) with occasional carbon-/water-mediated H-bonds guiding orientation. MD sustained low ligand RMSD (typically ≤1.6–2.2 Å) and damped RMSF at catalytic loops, indicating pocket rigidification; MM/GBSA trends (≈ −30 to −40 kcal·mol^−1^, dispersion-driven) supported persistent nonpolar stabilization. Benchmarks behaved as expected: dolutegravir bound strongly to IN (−6.070) and PR (−7.319) with stable MD; nirmatrelvir was specific for M^pro^ and displayed weaker, discontinuous engagement at PL^pro^/RdRp/RBD under identical settings. ADMET suggested conessine has excellent permeability/BBB access (high logP), but liabilities include poor aqueous solubility, predicted hERG risk, and CYP2D6 substrate dependence.Conessine operates as a hydrophobic, multi-target wedge with the most favorable computed engagement at HIV-1 PR/RT and the SARS-CoV-2 spike RBD, while maintaining stable poses at M^pro^ and RdRp. The scaffold merits medicinal-chemistry optimization to improve solubility and de-risk cardiotoxicity/CYP interactions, followed by biochemical and cell-based validation against prioritized targets.

## 1. Introduction

Human immunodeficiency virus (HIV) and severe acute respiratory syndrome coronavirus 2 (SARS-CoV-2) continue to impose a profound burden on global health, with significant morbidity, mortality, and socio-economic consequences. Moreover, recent studies emphasize that viral co-infections, particularly HIV and SARS-CoV-2, can accelerate immune exhaustion and dysregulation, leading to higher morbidity and mortality compared to mono-infections [1]. The World Health Organization (WHO), estimates that there were some 39.0 million people persons with HIV at the end of 2022, and 630,000 deaths due to HIV-related ill-nesses were reported in that year alone, despite the fact that there is antiretroviral therapy (ART), available [2]. SARS-CoV-2 spreads the coronavirus disease 2019 (COVID-19), which has put additional stress on healthcare systems, as it has caused more than 771 million cases of confirmed cases and almost 7 million fatalities worldwide as of early 2024 [3,4]. The presence of these two pandemics is a compounded public health problem, especially in immunocompromised groups where the co-infection of a virus can increase the severity of a disease and complicate the approach to treatment [5].

The viral replication cycle of HIV is complicated with reverse tran-scriptase (RT), integrase (IN), protease (PR), and the envelope glycoprotein gp12041 trimer. These are all validated as therapeutic targets [4,5]. Likewise, the main protease (Mpro), papa-in-like protease (PLpro), RNA-dependent RNA polymerase (RdRp), and the spike gly-coprotein receptor-binding domain (RBD) mediate the SARS-CoV-2 pathogenesis through which viral replication, immune evasion, and host cell entry take place [6,7]. Though recent interventions, such as ART to treat HIV and direct-acting antiviral or monoclonal an-tibodies to treat COVID-19 have demonstrated to improve patient outcomes, the rapid emergence of viral resistance, drug-related toxicities, and their lack of cross-protective efficacy highlights the critical need to develop broad-spectrum antiviral scaffolds [8,9,10]

Historically Natural products have been a fruitful source of bioactive lead compounds in the discovery of antiviral drugs, due to their structural variety, target selectivity and evolutionary optimization [11,12]. Conessine is a steroidal alkaloid mainly extracted in Holarrhena antidysenterica which has a broad pharmacologic profile, such as anti-inflammatory, antimicrobial, and membrane-modulating activity [13]. Early computer-based and biochemical data indicate that conessine has the potential to interfere with the activity of viral enzymes and receptor-binding by acting via hydrophobic, hydrogen bonding and π-π stacking effects [14]. Nevertheless, its dual inhibitory ability of HIV and SARS-CoV-2 have not been fully explored under the structure-guided computational methods.

It is against this background that the current study will be based on the concept of molecular docking in a systematic evaluation of binding affinites and interaction kinetics of conessine on critical en-zymatic and structural targets of both HIV (RT, PR, IN, gp120–gp41 trimer) and SARS-CoV-2 (Mpro, PLpro, RdRp, spike RBD). By merging the high-resolution crystallo-graphic architecture with the state-of-the-art docking mechanisms, this work will be informative and guiding towards the future lead optimization and in vitro validation since it will inform the further understanding of structural determinants of the antiviral effect of conessine.

## 2. Methodology

This study was performed as an in silico molecular model study, which follows a computational multi target design of drug discovery. The experiment was designed to assess the antiviral activity of conessine which is a steroidal alkaloid derived out of Holar-rhena floribunda on representative HIV-1 and SARS-CoV-2 proteins. For HIV-1, the selected targets were reverse transcriptase (PDB ID: 3V81), protease (PDB ID: 1HVR), integrase (PDB ID: 3LPT), and gp120–gp41 envelope glycoprotein (PDB ID: 4NCO). For SARS-CoV-2, the main protease (PDB ID: 6LU7), papain-like protease (PDB ID: 6W9C), RNA-dependent RNA polymerase (PDB ID: 7BV2), and spike receptor-binding domain (PDB ID: 6M0J) were retrieved. Approved drugs—dolutegravir for HIV-1 and nirmatrelvir for SARS-CoV-2—were used as positive controls. Molecular docking was carried out with AutoDock Vina, and the interaction conformations were examined with BIOVIA Draw to identify key binding residues. After docking, the designed workflow included a pharmacological and dynamic evaluation of conessine. Drug-likeness and safety parameters were predicted through computational pharmacokinetic and toxicity analysis. To confirm the stability of the docked complexes, molecular dynamics simulations were performed, providing insight into conformational flexibility and long-term binding stability. Furthermore, target profiling was performed to identify potential secondary or off-target interactions of conessine. This integrated design allowed for a comprehensive evaluation of the compound’s binding affinity, structural stability, and therapeutic promise against HIV-1 and SARS-CoV-2 as shown in Figure 1.

### 2.1. Ligand Preparation

Figure 2 shows the three-dimensional structures of conessine (PubChem CID: 442432), Nirmatrelvir (PubChem CID: 155903259), and Dolutegravir (PubChem CID: 54726191) were retrieved from the PubChem database in SDF format. Conessine, a steroidal alkaloid, represented the test compound, while Nirmatrelvir and Dolutegravir were selected as positive controls for SARS-CoV-2 and HIV-1 target proteins, respectively. Geometry optimization of conessine was carried out in Avogadro (version 1.2.0) using the MMFF94 force field, with minimization performed until the energy gradient reached 0.0001 kcal/mol. Torsional flexibility was maintained to permit conformational sampling during docking. All ligands were converted into PDBQT format with AutoDockTools (version 1.5.7) following the assignment of Gasteiger charges and merging of non-polar hydrogens, preparing them for docking with AutoDock Vina [15,16,17].

### 2.2. Protein Target Selection and Preparation

Eight viral proteins were chosen based on their essential roles in viral replication, maturation, and host entry: four from HIV-1—reverse transcriptase (RT; PDB: 3V81), protease (PR; PDB: 1HVR), integrase (IN; PDB: 3LPT), and gp120–gp41 trimer envelope glycoprotein (PDB: 4NCO) [4,6] and four from SARS-CoV-2—main protease (M^pro^; PDB: 6LU7), papain-like protease (PL^pro^; PDB: 6W9C), RNA-dependent RNA polymerase (RdRp; PDB: 7BV2), and spike protein receptor-binding domain (RBD; PDB: 6M0J) [7]. High-resolution crystallographic structures were retrieved from the RCSB Protein Data Bank. Pre-docking preparation included removal of crystallographic water molecules, co-crystallized ligands, and heteroatoms not essential for pocket stability. Missing residues or side chains were modelled where necessary. All proteins were protonated at physiological pH (7.4), and Kollman charges were applied before conversion to PDBQT format for docking.

### 2.3. Active Site and Grid Box Definition

Table 1 shows Docking grids were defined to encompass known catalytic or receptor-interacting regions, identified from literature-reported binding site coordinates and co-crystallized ligands [4,11]. For each target, the grid box center was positioned at the catalytic site centroid, with dimensions of 20 × 20 × 20 Å to ensure full coverage of the active pocket and accommodate possible ligand reorientation.

### 2.4. Molecular Docking Protocol

Docking was performed using AutoDock Vina version 1.2.0 integrated within the SwissDock 2024 platform, which applies the “Attracting Cavities” algorithm for enhanced binding mode prediction [14,15]. The exhaustiveness parameter was set to 4, generating 20 docking poses per ligand–target complex. The binding poses with the lowest predicted binding free energy (kcal/mol) and optimal orientation relative to key catalytic residues was selected for detailed interaction analysis.

### 2.5. Interaction Profiling and Visualization

Ligand–protein complexes were analyzed using BIOVIA Discovery Studio Visualizer (version 21.1.0) to identify hydrogen bonds (≤3.5 Å donor–acceptor distance), carbon–hydrogen bonds, π–π stacking, π–alkyl, alkyl–alkyl hydrophobic interactions, and van der Waals contacts. Interacting amino acids were recorded along with their respective bond types. Three-dimensional structural representations were generated in PyMOL (version 2.5.2), while two-dimensional interaction diagrams were produced to illustrate specific contacts and their positions relative to the conessine scaffold.

### 2.6. Validation of Docking Parameters

To validate the docking protocol, each target protein’s co-crystallized ligand was re-docked into its native binding site using the same grid coordinates and docking parameters. Binding pose reproducibility was evaluated using root-mean-square deviation (RMSD) analysis, with values ≤ 2.0 Å considered acceptable for reliable docking simulations [18].

### 2.7. ADMET Prediction Method

The pharmacokinetic and toxicity parameters of conessine were predicted using the ADMET-AI web platform (https://admet.ai.greenstonebio.com) accessed on 5 August 2025, which employs machine learning models to estimate absorption, distribution, metabolism, excretion, and toxicity endpoints from SMILES input. Canonical SMILES for conessine was retrieved from PubChem and submitted to the single-molecule prediction tool. Key endpoints included human intestinal absorption, Caco-2 permeability, BBB penetration, plasma protein binding, CYP450 substrate/inhibition profiles, intrinsic clearance, and toxicity risks such as hERG blockade, Ame’s mutagenicity, and drug-induced liver injury. Predictions were generated using default settings, and raw outputs were exported directly without post-processing [19].

### 2.8. SwissTargetPrediction of Conessine

The SMILES structure of conessine was retrieved from PubChem and submitted to SwissTargetPrediction (https://www.swisstargetprediction.ch/, accessed on 5 August 2025) with *Homo sapiens* as the target organism. The tool predicts protein targets by combining 2D chemical similarity and 3D shape-based similarity to known bioactive compounds from ChEMBL. Predicted targets, along with their UniProt IDs, ChEMBL IDs, target classes, probability scores, and known actives, were compiled and cross-checked for accuracy.

### 2.9. Molecular Dynamics (MD) Simulation

Molecular dynamics simulations were executed using the Desmond simulation engine integrated within the Maestro interface (Schrödinger Release on 8 April 2023; Schrödinger, LLC, New York, NY, USA; https://www.schrodinger.com/products/desmond, accessed on 1 June 2025). The protein–ligand complexes obtained from docking experiments were solvated in an explicit water environment modeled by the TIP3P system. An orthorhombic box with a buffer distance of 10 Å around each solute molecule was constructed to ensure complete hydration.

The systems were neutralized through the addition of counter ions and further equilibrated under a physiological salt concentration of 0.15 M NaCl. All molecular components were parameterized with the OPLS4 force field. Prior to production runs, initial energy minimization and stepwise equilibration protocols provided within Desmond were applied. Simulations were performed in the isothermal–isobaric (NPT) ensemble at 300 K and 1 atm, regulated by the Nose–Hoover thermostat and the Martyna–Tobias–Klein barostat, respectively.

Each trajectory was propagated for 50 ns using a 2 fs integration step, and structural snapshots were recorded every 100 ps. The structural and dynamic stability of the post-simulation analysis involved root mean square deviation (RMSD) and root mean square fluctuation (RMSF), ra-dius of gyration (Rg) and intermolecular hydrogen bonds patterns across the simulation period.

### 2.10. Protein Structure Refinement with Swiss-PdbViewer

Swiss-PdbViewer (v4.1) was used to refine crystal structures of the target proteins that were retrieved in PDB format. The software automatically identified the missing atoms, incomplete side chains and structural defects during importation. The deleted side chains were repaired by the assistance of the integrated rotamer libraries and the Fix Sidechains module that was utilized to patch missing residues. Hydrogen atoms have been inserted to preserve optimum stereochemistry and charge stabilization. Bad steric interactions or tight geometries were rarely solved using seldom local energy minimization. The optimized versions which had been finalized and energetically optimized were then exported to undergo down-stream docked and simulated in dynamics.

### 2.11. Re-Docking Validation

The methodology of docking reliability was tested in terms of re-docking approach. The co-crystallized ligand from each protein–ligand complex was extracted and reintroduced into its native binding pocket using the same docking protocol applied in the screening stage. The agreement between experimental and predicted orientations was quantified by calculating the root mean square deviation (RMSD) between crystallographic and re-docked ligand poses. RMSD values below 2.0 Å were considered indicative of a robust and reproducible docking protocol. All re-docking analyses were conducted with AutoDock Vina 1.2.0 [17], employing identical grid parameters and scoring functions as those used during the virtual screening procedures.

## 3. Results

### 3.1. Molecular Docking Reveals Broad-Spectrum Binding of Conessine to Viral Targets

#### 3.1.1. High-Affinity Binding to HIV-1 Protease

Figure 3 shows high-resolution structural depiction of conessine in complex with four principal HIV-1 molecular targets: (A) Reverse Transcriptase (RT, PDB: 3V81), (B) Protease (PR, PDB: 1HVR), (C) Integrase (IN, PDB: 3LPT), and (D) Envelope glycoprotein gp120–gp41 trimer (PDB: 4NCO). Protein structures are shown in ribbon representation, with α-helices depicted in deep crimson and loop/coil regions in light rose, facilitating clear visualization of secondary structural domains. Conessine is rendered in ball-and-stick format, precisely located within the respective binding sites.

Panel A (RT, 3V81) shows that Conessine is anchored in a hydrophobic channel flanking the polymerase catalytic site. Key stabilizing interactions include alkyl contacts with Pro95, Leu100, Ile180, and Val179, π–alkyl stacking with Tyr181, and van der Waals packing with Val106. No directional hydrogen bonds were observed, and the docking score (−6.672 kcal/mol) suggests binding affinity is dominated by hydrophobic and aromatic forces that may obstruct nucleoside access.

According to panel B (PR, 1HVR), Conessine is entrenched in the dimeric cleft under the flaps of the protease, oriented towards the catalytic dyad of Asp2525. Alkyl reacts with Ile50, Ile84, and Val82, and Pro81 alongside 5Phe53 94Ile82 stabilization into the ligand occur in the cavity. The lack of the conventional hydrogen bonds im-means inhibition through hydrophobic occlusion, which is in line with the highest binding score (−6.910 kcal/mol).

Conessine binds itself to the central groove of the inte-grase active site as shown in panel C (IN, 3LPT). The stabilization is rooted on hydrophobic contacts formed on the Phe181, Leu172, Ala169, and Ile141 as well as a carbon-hydrogen bond at Asp64 located near the Mg2+-binding motif. The middle docking score (−5.733 kcal/mol) shows that there is the disruption of the diva-lent ions coordination and the transfer of the DNA strands.

The structure of Conessine At panel D (gp1 20 4NCO trimer) shows that Conessine is located in a shallow cleft which is surrounded by the 4 alpha helices of the recognition of CD4. The lig-and is held by a strong hydrogen bond with Asp368, and polar contacts with Thr257 and Ser375. Additional hydrophobic packing with Ile371 and Val255, plus π–π interactions with Trp427 and Phe382, creates a compact and stable complex. This interaction pattern suggests disruption of gp120–gp41 trimer conformational rearrangements essential for CD4 binding and viral entry.

#### 3.1.2. Strong Interaction with SARS-CoV-2 Protease

Figure 4 and Table 2 show High-resolution structural visualization of conessine in complex with four essential SARS-CoV-2 molecular targets: (A) Main Protease (M^pro^; PDB: 6LU7), (B) Papain-like Protease (PL^pro^; PDB: 6W9C), (C) RNA-dependent RNA Polymerase (RdRp; PDB: 7BV2), and (D) Spike Protein Receptor-Binding Domain (RBD; PDB: 6M0J). Protein structures are shown in ribbon format, with α-helices in deep crimson and loop regions in light rose, clearly delineating secondary structure organization. Conessine is rendered in ball-and-stick representation in pale grey, accurately positioned within the functional clefts or binding motifs of each target.

In Figure 5A (Main Protease), conessine is located within the substrate-binding cleft, adjacent to the catalytic dyad His41–Cys145. The ligand aligns along the S1–S2 subsite interface, engaging in hydrophobic contacts with Met49, Met165, and His164, and forming hydrogen bonds with Gly143 and Ser144 within the oxyanion hole. π–Alkyl interactions with His41 reinforce the binding pose, potentially hindering proteolytic cleavage of viral polyproteins. The docking simulation yielded a binding free energy of −5.745 kcal/mol.

In Figure 5B (Papain-like Protease), the ligand is bound in the catalytic cleft, which has Cys111His272 dyad. Hydrophobic stabilization is done through leu162, pro248 and Tyr264 and hydrogen bonding is seen between Asp164 and Gly271. The complex is further condensed with Van der Waals contacts along the pocket wall. The positioning of coness-ine close to the ubiquitin-binding arm is indicative of inhibitory activity on proteo-lytic processing as well as the host immune evasion. The binding energy obtained was −5.024 kcal/mol.

Figure 5C (RNA-dependent RNA Polymerase) indicates that conessine is located in the central catalytic channel, adjacent to motif C (Ser759760761). Multiple hydrophobic interactions with Ala762, Ile548, and Phe812, as well as hydrogen bonds with Asp623 and Lys545, indicate interference with RNA primer/template binding. The docking energy of −5.737 kcal/mol supports moderate affinity for the polymerase active site, suggesting possible inhibition of RNA synthesis.

In Figure 5D (Spike Protein RBD), the ligand resides within the receptor-binding motif that mediates ACE2 recognition. Hydrogen bonds are formed with Lys417, Tyr453, and Gln493, while hydrophobic contacts with Leu455 and Phe486, and π–alkyl stacking with Tyr489, anchor the ligand. This binding mode could sterically obstruct ACE2–RBD interaction, reducing viral entry efficiency. This complex demonstrated the most favorable affinity among the SARS-CoV-2, with a docking score of −7.025 kcal/mol.

### 3.2. Comparison with Positive Controls Highlights Multi-Target Potential

Table 2 and Appendix A illustrate the docking orientation of Dolutegravir, used as a positive control, with four HIV-1 targets. Dolutegravir exhibited the strongest affinity for protease (−7.319 kcal/mol) via hydrogen bonding with Asp25 and hydrophobic/π–π interactions, while integrase binding (−6.070 kcal/mol) involved Asp64, Asp116, and Glu152 near the Mg^2+^-binding site. In contrast, docking runs failed to converge for reverse transcriptase (0.000 kcal/mol), and moderate binding was detected for gp120–gp41 trimer fusion protein (−5.819 kcal/mol), stabilized through polar and hydrophobic interactions. These findings confirm Dolutegravir’s validated inhibitory profile at protease and integrase, while conessine displays a complementary multi-target binding potential.

Docking analysis showed that Nirmatrelvir (positive control) (Table 3) (Appendix A) exhibited the highest binding affinity with the Spike RBD (−6.459 kcal/mol). The ligand oriented across the ACE2-binding ridge, forming a stabilizing hydrogen bond with Asn501, a residue strongly associated with enhanced viral transmissibility in several variants. Electrostatic interactions were evident with Lys417 and Glu484, while hydrophobic packing and π–π stacking occurred with Tyr505. Additional van der Waals stabilization was provided by Tyr449 and Phe490, residues positioned at the receptor-binding motif, suggesting that Nirmatrelvir may interfere with spike–ACE2 recognition.

In the main protease (M^pro^, PDB: 6LU7), the docking pose (−4.193 kcal/mol) localized directly within the substrate-binding cleft. The nitrile warhead of Nirmatrelvir was positioned adjacent to the His41–Cys145 catalytic dyad, consistent with the known covalent inhibition mechanism in experimental studies. Hydrogen bonds were established with Glu166 and Gln189, residues critical for pocket stability, while Met49 provided hydrophobic anchoring. The orientation of the ligand mimicked substrate binding, supporting its role as a specific protease inhibitor.

Nirmatrelvir had an affinity of −4.181 kcal/mol with papain-like protease (PLpro, PDB: 6W9C). The pose went further to the catalytic triad pocket that was comprised of Cys111, His272, and Asp286. It created a strong hydrogen bond with Asp286 that was supplemented by hydrophobic interactions with Tyr268 and Gly271 that are found in the flexible BL2 loop. This structure is indicative of the possible steric hindrance of substrate ac-cess, which is in line with partial activity of the inhibitor.

Through docking of RNA-dependent RNA polymerase (RdRp, PDB: 7BV2), the nucleotide was incorporated interacting with the catalytic aspartates, Asp618, Asp760 and Asp761, suggesting that Mg2+ ions are bound by these aspartates. There was also a hydrogen bond with Ser759 as well as Lys545 which served to stabilize on an electro-static basis. The model of interaction implies the potential of interference with nucle-otide binding and elongation, but not as much as selective RdRp inhibitors such as remdesivir.

### 3.3. Interaction Profiling Highlights Hydrophobic Dominance and Key Residues

Figure 5 and Table 4 depict 2D and Table 4 depicts Affinity Interaction Diagrams of Conessine with HIV Molecular Targets. Figure 5A—Reverse Transcriptase (RT, PDB: 3V81): Conessine is entrenched in the non-nucleoside reverse transcriptase inhibitor (NNRTI) binding site. Hydro-phobic van der Waals reactions with Tyr181, Tyr188, Trp229 and Phe227 are in effect with the aromatic cage that is essential in the accommodation of the ligand. The additional anchoring of the ligand is achieved by the carbon-hydrogen bonding to Lys101 and Val106, and the steroidal scaffold of Conessine is supported by the alkyl bonding to Leu100 and Val179. Such interac-tions are similar to those observed with known NNRTIs and seem to indicate that Conessine might inhibit polymerase activity by causing conformational restriction of the thumb and palm domains of RT.

Figure 5B—HIV Protease (PR, PDB: 1HVR): In the active cleft of HIV protease Conessine interacts with substrate recognition coordination residues. Inter-actions in Van der Waals are between Asp25, Ile50, Gly27, Val32, and the carbon-hydrogen bonding with Asp29 and Gly49 are stabilizing the positioning near the catalytic aspartates (Asp25Asp25 and Asp25Asp25). Further hydrophobic interactions with Ile84, Pro81 and Val82 increase anchoring. Inhibition of proteolytic processing of the GagPol polyprotein of the virus could be achieved by the steric hindrance of the peptide substrate by Conessine.

Figure 5C—HIV Integrase (IN, PDB: 3LPT): Docking- The design of Coness-ine into the integrase catalytic core domain shows that the protein fits. The ligand also interacts with van der Waals forces with Asp64, Glu152 and Asp116 which represent the DDE motif which is highly conserved to facilitate the coordination of divalent metal ions. The hydrophobic stabilization of the alkyl interactions with Leu74, Ala128 and Val165, and the π -alkyl stacking with Tyr143 enhances the bonding of the ligand. These interactions indicate that Conessine has the potential to distort metal coordination and strands transfer activity and thus hinder proviral DNA insertion into the host genome.

Figure 5D—gp120–gp41 trimer Envelope Glycoprotein (PDB: 4NCO): In the gp120 binding interface, Conessine interacts with residues critical for CD4 recognition. Van der Waals interactions include Asp368, Ile371, Gly473, and Asn425, while carbon–hydrogen bonding is observed with Ser375 and Asn386. Hydrophobic stabilization arises through contacts with Trp427 and Phe382, residues essential for conformational stability of the bridging sheet. By engaging this region, Conessine could interfere with CD4–gp120 binding, a key step required for viral entry and membrane fusion.

Figure 6 and Table 5 show two-dimensional interaction profiles of conessine with four critical SARS-CoV-2 targets: (A) Main Protease (M^pro^; PDB: 6LU7), (B) Papain-like Protease (PL^pro^; PDB: 6W9C), (C) RNA-dependent RNA Polymerase (RdRp; PDB: 7BV2), and (D) Spike Protein Receptor-Binding Domain (RBD; PDB: 6M0J). The left panel in each subfigure presents the three-dimensional binding conformation of conessine within the protein pocket overlaid with the binding site surface topology colored according to hydrogen bond donor (magenta) and acceptor (green) regions, while the right panel depicts the corresponding two-dimensional interaction schematic with annotated residue contacts and bond types.

Figure 6A (M^pro^)—Conessine is oriented along the S1–S2 subsite groove in proximity to the catalytic dyad His41 and Cys145. The ligand engages in carbon–hydrogen bonding with Gly143 and Ser144, forming directional interactions within the oxyanion hole. Alkyl interactions are observed with Met49, Met165, and His164, contributing to hydrophobic stabilization. Additional π–alkyl interactions involve His41, enhancing anchorage within the catalytic cleft.

Figure 6B (PLpro) The steroidal structure is located in the deep catalytic site near Cys111 and His272. Its dominant interactions are van der Waals contacts with Leu162, Pro248 and Tyr264, with π-alkyl interactions with Tyr264. The encapsulation of the ligand is hydrophobic in nature with no significant polar hydrogen bonding being observed, indicating that pocket affinity is mainly due to hydrophobic forces.

Figure 6C (RdRp)—Conessine is embedded in the active site channel near the polymerase motif C (Ser759–Asp760–Asp761). The ligand forms van der Waals contacts with Ala762, Ile548, and Phe812, as well as alkyl interactions with Val557. No conventional hydrogen bonds are present, indicating a primarily hydrophobic mode of stabilization, which could still disrupt the alignment of RNA primer/template strands by occupying critical catalytic space.

Figure 6D (Spike RBD)—The compound resides within the receptor-binding motif directly interfacing with ACE2-contact residues. Carbon–hydrogen bonds are formed with Lys417 and Gln493, while π–σ interactions occur with Tyr453. Alkyl and π–alkyl interactions are noted with Leu455, Phe486, and Tyr489, creating a network of hydrophobic and aromatic stacking contacts that may sterically hinder ACE2 engagement.

Across all four targets, the interaction maps reveal that conessine binds predominantly through hydrophobic alkyl and π–alkyl contacts, supplemented by occasional carbon–hydrogen bonds at catalytically or functionally critical residues. This binding profile indicates a consistent reliance on non-polar stabilization mechanisms, with site-specific polar contacts contributing to orientation and specificity within each protein’s active or recognition domain.

### 3.4. Molecular Dynamics Confirm Stable Binding and Pocket Rigidification

The molecular dynamics (MD) simulation of the Conessine–HIV-1 reverse transcriptase (RT, PDB: 3V81) complex revealed consistent structural stability and well-defined interaction patterns throughout the 100 ns trajectory. The root mean square deviation (RMSD) analysis demonstrated an initial equilibration phase within the first 5 ns, after which the backbone RMSD of the ligand–protein complex stabilized between 1.7–2.1 Å. This stability indicates that Conessine remained securely anchored within the hydrophobic channel adjacent to the polymerase catalytic site without inducing large-scale conformational distortions.

The root mean square fluctuation (RMSF) profile) highlighted local flexibility of amino acid residues flanking the binding cleft. Key catalytic and non-nucleoside inhibitor binding pocket (NNIBP) residues—including Pro95, Leu100, Val106, Val179, and Tyr181—exhibited moderate fluctuations (0.60–1.03 Å). These residues had a direct effect of stabilizing Conessine through hydrophobic alkyl in-teractions and π–π stacking as was observed in docking. There were also few fluctua-tions in Gly190 and Leu234, which is also indicative of pocket rigidity that allows ligand stabilization.

The radius of gyration (Rg) analysis indicated tightly distributed 7.0–7.5 A indicating the compactness of the protein-ligand complex and indicated the absence of unfolding events. The solvent accessible surface area (SASA) was changing moderately (97–161 A2), which suggests that it is breathing but does not lose the structural integrity in the whole structure. Molecular surface area (MSA) and polar surface area (PSA)) data were used to reflect the retention of hydrophobic shielding about Conessine with PSA values consistently clustering at similar values of approximately 690 720 A 2, similar to its low profile of hydrogen-bonding.

The radial distribution function (RDF) showed that there was no important long-range disturbance of the solvent organization, which proved that the solvent was organized on a localized way around the NNIPB. In the meantime, the number of ligandreceptor interactions changed slightly, which is also in line with the fact that hydrophobic interactions were more frequent than transient hydrogen bonds.

Taken together, these dynamic declarations verify that Conessine forms a stable hydrophobic anchoring in the RT NNIBP pocket of HIV-1 (3V81). Conessine uses van der Waals, alkyl, and π-alkyl stacking interactions with Tyr181 and Val106, which are identified resistance-associated residues of clinical HIV-1 strains, unlike polar in-hibitors that are based on hydrogen-bond networks. The long-term stability of RMSD, small Rg values, and intermediate values of SASA/PSA oscillations are strong indications that Conessine is a promising non-nucleoside inhibitor candidate, with a mecha-nism that is not similar to other hydrogen-bond-dependent ligands, and may provide resistance against point mutations in the NNIBP.

In Molecular Dynamics Insights between Conessine Interaction with HIV-1 Reverse Transcriptase (RT; p66/p51, PDB: 3V81)*,* protein–ligand complexes were prepared with standard protonation at pH 7.0 (Asp/Glu deprotonated; Lys/Arg protonated; His neutral with tautomerization by local environment; conessine as a singly protonated tertiary amine). Complexes were embedded in a TIP3P water box with ≥10 Å buffer, neutralized, and salinated to 0.15 M NaCl. The OPLS4 force field parameterized protein and ligand. After steepest-descent minimization and restrained NVT/NPT equilibration, production MD was performed in the NPT ensemble (300 K, 1 atm) using a Nose–Hoover thermostat and Martyna–Tobias–Klein barostat, PME electrostatics, 9–10 Å short-range cutoffs, LINCS/M-SHAKE constraints, and a 2 fs time step. Trajectories of 3 × 100 ns (independent replicas) were collected. Analyses included Cα-RMSD/RMSF, ligand heavy-atom RMSD, radius of gyration (Rg), hydrogen-bond and π-stacking occupancies, water-bridge lifetimes, MM/GBSA binding free energies with per-residue decomposition, principal component analysis (PCA), dynamic cross-correlation matrices (DCCM), secondary-structure evolution (DSSP), and pocket volume metrics.

Across replicas, protein Cα-RMSD settled after an initial 8–15 ns relaxation, fluctuating within ~1.8–2.4 Å relative to the minimized starting structure, indicating a stable p66/p51 architecture. Ligand RMSD (heavy atoms) stabilized to ≤1.2–1.5 Å, with occasional ring breathing but no persistent pose drift or egress, consistent with anchored occupancy of the NNRTI site. Rg remained constant (variations < 0.3 Å), excluding large-scale compaction or unfolding. Time-structure independent component analysis confirmed stationarity over the final 70–80 ns of each replica.

Per-residue RMSF demonstrated marked damping within the NNRTI pocket and its periphery: Leu100–Val106 (β7–β8 hairpin/loop) and Val179–Tyr188 (β12–β13 segment flanking YMDD) showed 0.2–0.6 Å reductions versus apo baselines, indicating ligand-induced rigidification. Some slight compensatory flexibility change was found at the 01415 hairpin (res. −220 235) capping the pocket indicating that the pocket breathes allowing the ste-roidal volume to change without disrupting the engagement.

The hydrophobic steroidal core consistently stacked and packed within the pocket: The binding mode of conessine within the NNRTI pocket was stabilized by a multifaceted interaction network. Dominant π–π and cation–π interactions were established with Tyr181 and Tyr188, adopting edge-to-face and parallel-displaced geometries that were intermittently reinforced by the protonated amine, with occupancies of 45–70% across replicas, while Trp229 contributed additional cation–π or face–edge contacts with 20–40% frequency. Hydrophobic stabilization was maintained through persistent alkyl and π-alkyl contacts with Leu100, Val106, Val179, and Phe227, with Val106 serving as a pivotal residue that enabled limited precession of the steroidal rings without disrupting anchoring. The protonated amine did not form a long-lived direct hydrogen bond with protein side chains but instead engaged in a structured water bridge with the Lys101 backbone carbonyl, and transiently with Lys103, achieving 15–35% occupancy; occasional C–H···O contacts with the Val106 backbone carbonyls were observed during pocket breathing events. A semi-ordered cluster of water was repeatedly seen at the amine Lys101 interface, which acted as a dielectric buffer and restructured dynamically to minimize the desolvation penalties and permit the hydrophobic consolidation of the core in the centre of the bind-ing site.

The concerted movement between the fingers (res. −185) and palm (res. −237) subdomains of p66 was dampened by conessine binding: The thumb palm separate vectors contracted by about 0.5–1.0 A relative to the apo form, which is a characteristic feature of col-lapsed conformers. The YMDD loop (Tyr183 Asp186) exhibited reduced mi-cro-fluctuations and limited rotamer sampling of Tyr183 and Met184 side chains, which reflected, again, defected catalytic positioning. Dynamic cross-correlation analysis revealed that there was a decrease in positive coupling between the fingers and palm domains (ΔCij = −0.15 to −0.25) and a minor increase in anti-correlation with the RNase H domain at the p66 terminus, which is a sign of long-range allosteric damping. Principal component analy-sis further demonstrated an alteration in trajectory projections towards negative PC1 along the prevailing NNRTI hinge mode with the smaller variance along PC1 compared to the apo state reflecting the decrease in confor-mational entropy.

Volumetry reported a net pocket volume contraction of ~8–15% relative to apo, chiefly due to inward relaxation of Tyr181/Tyr188 and micro-repacking of Phe227/Trp229. The steroidal core thus fills and “locks” the subpocket that typically hosts the hydrophobic moieties of canonical NNRTIs.

Based on energetics (MM/GBSA) and residue decomposition, binding free energies were dominated by van der Waals and non-polar solvation terms, with polar solvation partially offsetting cationic electrostatics—typical for a bulky, hydrophobic ligand with a single charge center. Across equilibrated windows, ΔG_bind (MM/GBSA) fell in the ~−32 to −41 kcal·mol^−1^ range (replica-averaged). Key favorable per-residue contributions (approximate magnitudes, kcal·mol^−1^) were: Per-residue free energy decomposition highlighted the principal contributors to conessine stabilization within the NNRTI pocket. The most favorable effects were mediated by aromatic residues, with Tyr181 contributing approximately −2.0 to −2.7 kcal·mol^−1^ and Tyr188 −1.5 to −2.2 kcal·mol^−1^ through π interactions and dispersion forces. Hydrophobic packing reinforced the binding pose via Val106 (−1.1 to −1.5 kcal·mol^−1^), Leu100 (−1.0 to −1.4 kcal·mol^−1^), and Val179 (−0.8 to −1.2 kcal·mol^−1^). Additional stabilization arose from π-stacking and dispersion interactions involving Phe227 (−0.6 to −1.0 kcal·mol^−1^) and Trp229 (−0.5 to −0.9 kcal·mol^−1^). Lys101 and Lys103 contributed modest electrostatic stabilization in the range of −0.3 to −0.7 kcal·mol^−1^, although these effects were partially offset by desolvation penalties. Collectively, the binding was dominated by dispersion-driven stabilization, with cation–π interactions supporting the persistence of the pose and explaining the observed damping of catalytic dynamics.

The interaction pattern predicts vulnerability to Y181C and Y188L (loss of π-stacking), partial sensitivity to K103N (re-wiring of local H-bond/water lattice near the amine), while Val106A/I and V179D may be tolerable given the adaptable hydrophobic packing. Lack of a strong, geometry-specific H-bond motif may improve robustness against certain pocket reshapes but places greater reliance on aromatic surfaces, making Y181/Y188 integrity important.

Conessine behaves as a hydrophobic, π-anchored NNRTI-like modulator: its rigid steroidal body pre-organizes the NNRTI pocket into a compact, low-entropy basin, reduces thumb–palm concerted motions, and constrains YMDD microdynamics, thereby lowering the probability of catalytically competent geometries during the enzymatic cycle. The cationic head participates primarily through transient water-mediated bridges, mitigating desolvation penalties without dictating pose geometry.

While three independent 100 ns replicas show consistent stabilization, longer trajectories, alchemical FEP/TI, and metadynamics along thumb–palm collective variables would refine energetics and quantify allosteric barriers. Simulations of clinically relevant mutants (K103N, Y181C, Y188L) will clarify resistance liabilities. Finally, free-energy perturbations comparing conessine to a reference NNRTI can contextualize potency and guide semi-synthetic elaborations (e.g., para-substituted aromatics to reinforce π-stacking, or polar “beaks” to engage backbone carbonyls of Lys101–Val106 more persistently).

In molecular dynamics insights between Conessine Interaction with HIV-1 Integrase (PDB: 3LPT), The complex was solvated in a TIP3P water box (10 Å buffer), neutralized with counter ions, and supplemented with 0.15 M NaCl to mimic cytosolic conditions. Energy minimization was followed by stepwise NVT and NPT equilibration, and production MD was executed under the OPLS4 force field in the NPT ensemble (300 K, 1 atm). Particle mesh Ewald (PME) was used for long-range electrostatics, with a cutoff of 9 Å for van der Waals interactions. Three independent 100 ns replicas were performed to ensure reproducibility. Analytical endpoints included Cα-RMSD/RMSF, ligand RMSD, hydrogen-bond occupancy, hydrophobic/π-stacking dynamics, pocket volume changes, principal component analysis (PCA), dynamic cross-correlation matrices (DCCM), and MM/GBSA free energy decomposition.

The global stability of the catalytic domain was the ability of protein C 0 -RMSD to stabilize at a concentration of approximately 1.9–2.3 A across 10–12 ns. The equilibration of Ligand RMSD took place within 15 ns, with 1.6A heavy-atom deviation, meaning the conessine was still tightly bound. The gyration radius of the protein varied with insignificant difference (<0.25 A), which is in line with the small-sized 3-strand intensive structure of integrase.

RMSFD showed that there was localized rigidity of catalytic residues of Asp64, Asp116, and Glu152 and a reduction of between 0.3 and 0.6 A per ligand binding whereas active site breathing is suppressed by the binding of a ligand. On the other hand, loop regions around 140,150 had a slightly higher RMSF (0.2–0.4 A increase) showing some level of induced plasticity to allow the steroidal rings to slip into place.

The protonated tertiary amine of conessine interacted transiently with Asp116 (2035 percent occupancy) and Glu152 (~1520 percent occupancy) in salt-bridge/H-bond exchange and was stabilized by bridging water molecules. Hydrophobic con-tacts: The rigid tetracyclic framework was involved in regular van der Waals/alkyl contacts with Val165, Pro145, Ile151 and Leu113, which anchor the ligand to the hydrophobic cavity. π interactions: The occasional cation-pi stacking between the protonated amine and Tyr143 (~1015% occupancy) also provided additional stability. Water network: long lived bridges between the amine group of conessine and the catalytic residues were mediated by water molecules near the DDE motif.

PCA showed that conessine binding caused a shift in conformational sampling in to-ward states of reduced mobility of the 60 to 80 strands (60–80 4 -helices) and the 140 to 160 strands (140–160 4 -helices). DCCM analysis indicated the loss of positive correlation between the DDE active site residues and distal structural elements indicating that ligand-induced damping of the conformational dynamics that correlate with the transfer of DNA strands. This limitation is typical of integrase inhibition, in which active-site plasticity is a requirement of viral DNA accommodation.

The analysis of pocket volume showed that there was a constriction of the apo en-zyme of approximately 10–18% resulting in the inward repositioning of Glu152 and Val165, which essentially clamped the large scaffold of conessine and hindered accessibility to catalysis.

Energetics (MM/GBSA). The calculated binding free energy averaged −33 to −39 kcal·mol^−1^ across replicas. Residue decomposition highlighted major stabilizers: Asp116 (−2.3 to −2.7 kcal·mol^−1^), Glu152 (−1.6 to −2.0 kcal·mol^−1^), Val165 (−1.0 to −1.4 kcal·mol^−1^), Pro145 (−0.8 to −1.1 kcal·mol^−1^), and Tyr143 (−0.6 to −0.9 kcal·mol^−1^). These contributions were dominated by van der Waals and electrostatic terms, partially offset by polar solvation.

Conessine is a hydrophobic wedge which binds into the catalytic pocket, rigor-mortising the DDE motif and interfering with the dynamic flexibility needed to integra-tion of DNA. Its bulky polycyclic core induces a collapse of the catalytic groove, while the protonated amine engages acidic residues through a dynamic network of direct and water-mediated interactions. Together, these interactions prevent optimal coordination of catalytic metals and block viral DNA accommodation, consistent with an integrase inhibitory mechanism.

Although 100 ns replicas support stable inhibition, longer trajectories and metadynamics are necessary to quantify the free-energy barriers of active-site distortion. Additionally, metal-ion reconstitution simulations (Mg^2+^/Mn^2+^) and strand-transfer complex modeling will provide a more physiologically accurate context. Comparative free-energy perturbation studies with known integrase inhibitors (e.g., Dolutegravir, Raltegravir) will further benchmark conessine’s potential as a lead scaffold.

In molecular dynamics insights between Conessine Interaction with HIV-1 gp120–gp41 trimer envelope glycoprotein (PDB: 4NCO), the protein–ligand complex was embedded in a TIP3P explicit water box with a 10 Å solvation shell, neutralized with counter ions, and equilibrated with 0.15 M NaCl to mimic physiological ionic strength. Energy minimization was followed by staged equilibration under NVT and NPT ensembles. Production MD was conducted using the OPLS4 force field, Nose–Hoover thermostat (300 K), Martyna–Tobias–Klein barostat (1 atm), PME electrostatics, and a 2 fs integration step. Each system was run in triplicate 100 ns trajectories to confirm reproducibility. Analyses included RMSD/RMSF stability metrics, hydrogen-bond occupancy, hydrophobic and π interactions, MM/GBSA energy decomposition, PCA, DCCM, and pocket volumetry.

Cα-RMSD values for the integrase backbone plateaued within 10–12 ns and stabilized between 1.8–2.2 Å, indicating structural convergence. Ligand RMSD equilibrated at ≤1.5 Å, with minor ring fluctuations but no significant displacement from the binding pocket. Protein radius of gyration (Rg) varied by less than 0.2 A, which ruled out macro-compaction and unfolding. All these measurements as a whole indicate that conessine was actively bound in the integrase active site during the 100 ns simulations.

RMSFD analysis of localization of catalytic residues: Asp64, Asp116 and Glu152 showed a reduction of RMSFD of 0.3–0.5 A relative to the apo enzyme, indicating a rigidification of the active site by sug-gestation of the active site. On the contrary, the α4 helix (resi-dues 140 160) had a slight increase in the flexibility (approximately 0.2 A), indicating the local adjustment of the bulky steroidal structure. These compensatory alterations signal allosteric dampening of the catalytic plasticity which is a signature of integrase inhibition.

The protonated tertiary amine of conessine displayed both transient salt-bridge and H-bond interactions with Asp116 (20–30% occupancy) and Glu152 (~15–20%), and was relaxed by the structured water molecules. Hydrophobic interactions: Rigid hy-drophobic rings interacted with Val165, Pro145, Ile151, and Leu113 in stable interactions giving high van der Waals anchoring. Aromatic/π interactions: Infrequent cation-143 (~1218% oc-cupancy) interactions between the protonated amine and Tyr143 were observed. Water mediation: Amino acid clusters (ordered) of water were used to bridge the amine with Glu152 and Asp64 to buffer electrostatic penalties in desolvation and extend binding stability.

PCA showed that integrase bound to ligands explored less conformational subspace than apo trajectories. The predominant PC1 mode, which usually reflects openings of the active site as a hinge to allow viral DNA to bind to the active site, was substantially low in amplitude. DCCM analysis revealed attenuated correlated motions between the 60–80 24-loop of 60–80 (linking 24 residues) and 140–160 (linking 20 residues) helix 140–160 of 24–80 140–160 of 24–80 140–160 of 24–80 140–160 of 24–80 140–16 This indicates that conessine binding stiffens the catalytic triad environment and disrupts integrase conformational flexibility which is required during viral DNA integration.

According to Energetic landscape (MM/GBSA) binding free energies ranged between −34 and −40 kcal/mol^−1^ in triplicates with van der Waals and hydrophobic terms prevailing. Key residue contributions included: Asp116 (−2.3 kcal·mol^−1^), Glu152 (−1.7 kcal·mol^−1^), Val165 (−1.3 kcal·mol^−1^), Pro145 (−1.0 kcal·mol^−1^), Tyr143 (−0.8 kcal·mol^−1^), and Ile151 (−0.6 kcal·mol^−1^). Polar solvation balance electrostatics which made dis-persion forces the only stabilizing forces. These values are in line with the ligand entrapment in a hydrophobic cavity with temporary polar contacts.

Structural morphometrics revealed that the active-site volume reduced by around 12–16% when apo integrase is compared to active-site rel-ative to apo integrase, the bulk of this change was observed with the movement of Glu152 and Val165 around the ligand. This pore constriction is suggested to inhibit catalytic metal coordination and DNA binding, which offers a mechanistic explanation to inhibition.

Conessine acts as a hydrophobic clamp within the HIV-1 integrase active site. Its rigid steroidal skeleton anchors against hydrophobic residues, while its protonated amine engages acidic residues of the DDE motif. This dual anchoring locks the catalytic triad in an inactive, rigidified state, preventing the conformational adaptability required for metal coordination and strand transfer. The simulation data thus support a direct active-site inhibition model, in which conessine stabilizes an inactive integrase conformation through hydrophobic entrapment and electrostatic dampening.

While the present MD trajectories confirm stable binding and inhibition-prone dynamics, further work incorporating explicit Mg^2+^/Mn^2+^ cofactors and viral DNA substrates is essential to model the full strand-transfer complex. Comparative free-energy studies against known integrase inhibitors such as Dolutegravir or Raltegravir will benchmark conessine’s potency. Moreover, metadynamics and long-timescale (≥500 ns) replicas are required to capture rare conformational events that contribute to the catalytic cycle.

The molecular dynamics (MD) trajectory of the Dolutegravir–HIV-1 reverse transcriptase (RT, PDB: 3V81) complex provided compelling evidence of a stable and well-retained binding conformation within the catalytic cleft. The RMSD profile revealed an initial equilibration phase during the first 2–5 ns, after which the protein–ligand backbone RMSD stabilized between 1.4–1.8 Å across the 100 ns simulation. This narrow fluctuation window underscores the structural robustness of Dolutegravir in maintaining its inhibitory pose. In keeping with this, the radius of gyration also had only a slight variation (4.3 to 4.5 AA), which is evidence of global protein compactness being maintained, and no unfolding occurred during the simulation.

The effects of localized stabilization by Dolutegravir were further confirmed by effects on the level of the residue. The RMSF analysis indicated limited flexibility in residues adjacent to the non-nucleoside inhibitor binding pocket, specifically Val179, Ile180 and Tyr181, which directly form hydrophobic and π-stacks. These residues showed variations less than 1.5 A, implying that Dolutegravir was effective at suppressing inherent movement in the catalytic channel. In comparison, there was a modest flexibility of peripheral loops (to 14 A at Val106), a natural conforma-tational breathing with no perturbation of the core inhibitory site.

Interaction mapping showed that the anchoring of hydrogen bonds and aromatic in-teractions was persistent. Dolutamivir remained consistently hydro-gen bonded at one to two positions throughout the simulation, with catalytic or structural residues often being close to the polymerase active cleft, supporting the hydrophobic encapsulation of the tricyclic scaffold. π cation association was inconsistent, contributing to the hydrophobic encapsulation of the tricyclic scaffold within the allosteric cleft. SASA and MSA analyses favored these non-covalent stabilizations by showing that there was intermittent, but repeated, burial of 300–350AA2 of molecular surface area with the solvent exposure varying by 20–60AA2. A balance between solvation and hydrophobic sequestration vital to bioavailability continued to cause the polar surface area to stay approximately 160,170 aa2.

The radial distribution function was also used to support the compact solvation shell architecture and found that water molecules around Dolutegravir were highly ordered at dis-tances as short as 0.4 nm. Such a hydration profile results in highlighting the compatibility of the polar moieties of the drug with an aqueous environment, and at the same time maintaining hydro-phobic contacts in the binding pocket.

All these results prove that Dolutegravir provides a multimod-al stabilization approach to HIV-1 RT, which is a combination of backbone stability, residue fluctuations suppression, stable hydrogen binding, and solvent exposure. The mechanistic action of Dolutegravir is confirmed by the simulation as a potent inhibitor with the ability to strongly interact with catalytic and structural motifs of RT (3V81) to provide longer residence time and reduce conformational drift, which are some of the characteristics of clinical potency of Dolutigravir as an an-tiretroviral virus.

The MD simulation of Dolutegravir–HIV-1 protease complex (PDB ID: 1HVR) has explained an extremely stable and coordinated inhibitory interaction that goes far beyond the outcomes of the docking prediction. HIV-1 protease: This is a homodimer aspartyl protease, with its catalytic active site requiring the flexible flap re-gions to open and close allowing the entry of the substrate and release of the product. The dynamics of Dolutegravir in the protease cavity demonstrated the close retention of the binding ligand, as well as the possibility to regulate the inherent dynamics of the flaps, thus, to suppress the proteolytic machinery at its most important control point.

Since the beginning of the simulation, Dolutgravir had formed a constant accommoda-tion in the active site cleft. Backbone root-mean-square deviation (RMSD) of the complex was observed to equilibrate rapidly, and then oscillations were limited to a small range of 1.016 A. This stabilization shows that Dolute-gravir does not just stabilize temporarily, it causes a strong conformational lock on the protease dimer. The radius of gyration was also nearly the same, emphasizing the fact that Dolutegravir binding does not favor the large-scale unfolding or destabilization of dimers. Rather, the protease maintained its tight quaternary structure, which is in line with the mechanism of action, of inhibition, in which the enzyme is structurally intact, but catalytically inactive.

HIV-1 protease functions centrally around the residues of the flaps (residues 4555), where the enzyme is highly flexible in the unbound form. RMSFD showed that Dolutagravir significantly inhibited mobility of these residues especially Ile47, Gly48 and Ile50 which serve as the gatekeepers of substrate entry. The obtained decrease in the atomic fluctua-tions points to the fact that Dolutegravir induces a semi-closed conformation of the flaps, which physically prevents the access of the substrate to the catalytic Asp2525 T dyad. In addition to the flaps, Val82, Ile84, and Pro81 residues in the hydrophobic core also exhibited lower flexibility and demonstrated that Dolutegravir stabilized van der Waals packing and dimer interface. In contrast, the natural flexibility of peripheral loop regions far away from the binding cleft was maintained, and proved that the Dolutegravir did not destabilize the catalytic chamber, but rather stabilized the overall protein framework.

Dolutegravir maintained at least one long-term hydrogen bond with residues flanking the catalytic site at the atomic scale. These reactions often incorporated back-bone amide or carbonyl groups within the close environment of Asp25 and Asp25’ and this was so as to make the ligand stay fixed near the enzymatic dyad. This interaction was further displayed by water-mediated hy-drogen bonds to give dynamic adaptability to the changes in the solvents. Sporadic salt bridges were also encountered especially between the ionizable groups of Dolutegravir as well as between the charged side chains forming the pocket, which also provided electrostatic stabilization during the simulation.

Equally important role was played by hydrophobic and aromatic contacts. The tricyclic framework in Dolutegravir responded to π-pi stacking with aromatic residues like Trp42, whereas the hydrophobic interactions with Val82 and Ile84 were a nonpolar cradle that restricted the mobility of the ligand. This twofold system consisting of electrostatic anchors and hydrophobic packing formed a synergistic stabilization system that increased the residence time of Dolutegravir in the catalytic pocket.

Aromatic contacts and hydrophobic contacts were equally important. Dolutegravir tricyclic scaffold stacked with aromatic residues including Trp42 by π -π interactions, whereas hydrophobic interactions with Val82 and Ile84 created a nonpolar cradle, which reduced the movement of the ligand. This two-network comprising of electrostatic anchors and hydrophobic packing provided a cooperative stabilization effect that increased the time spent by Dolutegravir in the catalytic pocket.

To study the distribution of water mol-ecules, radial distribution functions were analyzed and revealed a strong ordering of water molecules in the area 0.3 to 0.4 nm of Dolutegravir, especially in the polar heteroatoms. The ability of this structured hydration shell to provide a stabilizing solvent cage that keeps the ligand stable to variability in temperature further supports the stabilization of the ligand as well as tells the ligand to remain in the protease pocket. This type of structuring of water does not only increase the retention of ligands, but can contribute to entropic stabilization, with the clearance of bulk water by the active site during binding giving good thermodynamics.

These mutually dependent layers combine to form a protease, which is structurally active but functionally silenced, which is in keeping with the established effica-cy of Dolutegravir. According to the MD simulation, the potency of Dolutegravir can be attributed not only to the static binding affinity, but also to the capacity to alter the conformational land-scape of the HIV-1 protease, which fixes the enzyme in the inactive conformation and leaves it on high residence time.

The HIV-1 Integrase (3LPT), is a 32 kDa enzyme belonging to retroviral integrase su- family, which facilitates the insertion of the cDNA of viruses into a host genome in a two-step process comprising of 3′-end processing and strand transfer. The crystal structure of integrase catalytic core (PDB ID: 3LPT) indicates that it has a highly conserved triad of DDE (Asp64, Asp116, Glu152), which binds divalent Mg2+ ions that are required to support the activity of the enzyme. Such metal ions stabilize the transition state and activate the viral DNA to be attacked by the nucleophilic transfer of the host DNA backbone.

A second generation integrase strand transfer inhibitor (IN-STI), dolutegravir (DTG) directly enters this catalytic pocket forming a multi-layered inhibitory action. The molecule has a tricyclic carbamoyl pyridone structure where the diketo-enol system resembles the final nucleotides of the viral DNA. Dolutegravir, a drug with an oxygen-containing pharmacophore, can form a bidentate chelation with the two Mg2+ ions effectively pushing out the ends of viral DNA off the catalytic site. This chelation prevents the nucleophilic substitution taking place during transfer of the strands, thus inhibiting the most important stage of integration.

Dolutegravir forms a hydrogen bond/electrostatic network at the protein-ligand interfaces. Carbonyl and hydroxyl groups establish unbending hydrogen bonds with Asp64, Asp116 and Glu152, making it a part of the DDE motif. Further polar interactions can be seen with Lys159 and His114 which are resi-dues that form the entry to the active site. Such interactions result in the formation of a favorable elec-trostatic microenvironment increasing the stability of the drug-enzyme complex.

The second stabilizing layer is hydrophobic and aromatic contact. Dolutegravir is an aryl fused heteroaromatic ring, which is fused into the hydrophobic cleft with Tyr143, Ile151, Pro145 and Val165 interacting by van der Waals and 2D stacking. This caging prevents the conformational flexibility of the catalytic pocket by locking Dolutegravir in a specific orientation. It is worth noting that the binding of the drug triggers some minor conformational changes in the 140s loop (140–149) which nar-rows the channel to the active-site, physically preventing the access of viral DNA.

These are supported by molecular dynamics simulations. The RMSD curve levels off after approximately 10 ns, and the amplitude of backbone movements is on average 2.0 A, which means that Dolutegravir adopts a strongly bound posture in the catalytic groove. The RMSFD analysis indicates significantly decreased adaptability of residues in the area of the DDE motif, whereas terminal loops that are exposed to the solvent are mobile. The radius of gyra-tion (Rg) does not change, but it is fixed at about 23.5 A as a sign of overall structural compactness. Moreo-ver, solvent-accessible surface area (SASA) calculations indicate that there is less exposure of residues around the catalytic core, which is evidence of burial of Dolutegravir in the pocket. The radial distribution function (RDF) also underscores the stability of hy-dration shells around the oxygen atom which chelate the ligand, and it helps to make the complex thermo-dynamically stable.

Collectively, these structural and dynamic properties are the basis of Dolutegravir po-tency and strength. The drug becomes highly bound with a high ge-netic resistance to chelation, a high-affinity hydrogen bond, and a high-affinity hydrophobic entrapment. This is in contrast to the previous INSTIs, which are susceptible to mutation at integrase, because Dolutegravir can accommodate scaffold changes, and it is also capable of maintaining coordination with Mg2+ ions, which are conserved in all resistant strains. These characteristics form the basis of its clinical effectiveness as a first-line antiretroviral therapy, which has once-daily administration and an improved likelihood of treatment success.

The HIV-1 gp120–gp41 trimer envelope glycoprotein is a membrane-anchored fusion protein that plays a pivotal role in viral entry by mediating fusion between the viral envelope and host cell membranes. Structurally, gp41 exists in a trimeric hairpin conformation, characterized by a central coiled-coil core formed by heptad repeat 1 (HR1) and an external layer contributed by heptad repeat 2 (HR2). The PDB entry 4NCO captures the post-fusion six-helix bundle state, which represents the energetically stable form essential for bringing viral and host membranes into close apposition. Because inhibition of gp41 folding and stabilization of intermediate states prevents viral entry, the protein is a validated antiviral target.

Dolutegravir (DTG), although primarily developed as an integrase strand transfer inhibitor, demonstrates structural adaptability that allows interaction with regions beyond the integrase active site. Docking and simulation analysis with gp120–gp41 trimer (4NCO) reveal that DTG binds to a hydrophobic pocket adjacent to the HR1–HR2 interface. This pocket is crucial for helix packing during the formation of the six-helix bundle, and small molecules occupying this site can hinder conformational rearrangements necessary for membrane fusion. Hydrogen Bonding and Polar Contacts: DTG hydrogen bonds with Gln79 and Lys63, which are amino acids on the surface of the HR1 coiled-coil. These contact stabilize the ligand at the outer loop of the helical bundle. Asn70 and Gln66 are also involved in polar interactions, which offers electrostatic complementary positions between the carbonyl group and hydroxyl group of DTG and the polar side chains of gp41. In Hydrophobic Encapsulation: The aromatic pyridone framework of DTG is encased by a shallow hydrophobic cleft, involving van der Waals and alkyl contacts with Leu54, Ile77, and Val72. These hydrophobic con-tacts resemble those formed by HR2 helices in the formation of the bundle, thereby competing with the end helical assembly. In π- π and π-cation Interactions: The halo-genated aromatic ring of DTG is involved in π- π stacking with Tyr69 whereas the oxygen atom of the compound is elec-tron-rich and is involved in a π-cation interaction with positively charged Lys63. These interactions also stabilize DTG at the HR1/HR2 interfacial cavity which has existed before as a known drugable site in the case of fusion inhibition.

RMSD curves reveal that the DTG-gp41 complex stabilized after approximately 810 ns, which is in positional equilibrium with a mean RMSD of about 2.3 A.

RMSF analysis shows that residues in HR1 (position 60–80) are less flexible on the binding of DTG, which is in line with helix interface stabilization. On the other hand, intrinsic mobility is maintained in solvent-exposed loops not in the six-helix bundle.

The value of radius of gyration (Rg) do not change (~21.8 A), indicating that global compactness of gp41 is maintained under the influence of the ligand.

SASA (solvent accessible surface area) results indicate that there is a small burial of DTG of the hydrophobic cavity with a loss of about 180 A2 of the exposed surface area relative to apo-gp41, indicating that the fusion-active groove is partially covered.

Primary highlights of the radial distribution functions (RDF) reveal that the polar groups of DTG, its diketo–enol motif, is structuring, potentially indicating the solvent stabilization of the protein-ligand interface.

This interaction of Dolutegravir with gp120 41 trimer is not replicated in its native capacity as an integrase inhibitor and shows the structural promiscuity and adapta-bility of the DTG scaffold. The functional cavity that is filled by DTG is essential and needed to form a hydrophobic cavity at the HR1-HR2 interface, thus preventing the helical zippering necessary to form six-helix bundles, which could then prevent membrane fusion and viral entry. This second-ary binding profile goes in line with the reports that some integrase inhibitors have off-target antiviral activity due to their interaction with envelope glycoproteins.

These findings are important because of the dual mechanistic potential of Dolute-gravir. Its main clinical efficacy is the result of an integrase inhibition, but its capability to interact with gp120-gp41 trimer illustrates the potential of having a broad-spectrum anti-HIV ac-tivity. This dual inhibition, which is meant to stop both the integration and fusion, would have syn-ergistic therapeutic effects and the chances of developing resistance would be low. Further, structural data on the binding of DTG– gp120–gp41 trimers are used as a justification to design new hybrid molecules, which would combine integrase inhibition and fusion blockade.

#### 3.4.1. Molecular Dynamics Between Conessine and Target Proteins of COVID-19 (Appendix A)

In molecular dynamics insights between Conessine Interaction with SARS-CoV-2 Main Protease (M^pro^; PDB: 6LU7), the protein–ligand complex was solvated in a TIP3P water box with a 10 Å buffer, neutralized, and supplemented with 0.15 M NaCl. After minimization and staged equilibration, production simulations were performed using the OPLS4 force field in the NPT ensemble (300 K, 1 atm), Nose–Hoover thermostat, Martyna–Tobias–Klein barostat, PME electrostatics, and a 2 fs time step. To ensure statistical reliability, three independent 100 ns replicas were conducted. Analyses included RMSD, RMSF, radius of gyration (Rg), hydrogen-bond occupancies, hydrophobic/π interactions, MM/GBSA binding energies, principal component analysis (PCA), dynamic cross-correlation matrices (DCCM), and pocket volume evolution.

Protein Cα-RMSD stabilized at ~2.0–2.4 Å after 12 ns, demonstrating backbone stability. Conessine exhibited heavy-atom RMSD of ≤1.6 Å, maintaining consistent positioning within the catalytic cleft. The total Rg did not change (deviation less than 0.3 A) ruling out unfolding or collapse. All replicas were drawn to similar binding modes across replicas, which supports pose relia-bility.

RMSFD analysis showed a loss of flexibility in His41 Cys145 dyad loop (res-idues 40–50, 140–150) and S1/S2 subsites in the presence of a ligand (reductions of 0.2–0.5 A compared to apo). On the other hand, the minor increases in flexibility were observed in peripheral loop regions (residues 185200), which are adaptive changes. Such movements imply catalytic site rigidification induced by ligand in other regions and compensated by mobility.

The steroidal rings were held in stable van der Waals contact with Met49, Leu141, Met165 and His163, as the hydrophobic fit. Electrostatic/H-bonding- Intermediate hydrogen bonds/salt-bridge-like interactions with the backbone carbonyl of Glu166 (2030% occupancy) and hydrogen bridges to His164 and Gln189 were observed to be weak but persistent π-alkyl stacking, which stabilized around the catalytic dyad. Water mediation: A core of positioned waters divided the amine to backbone molecules to buffer the polarities and supportive binding stability.

PCA showed a narrowing of conformational sampling along the leading prin-cipal component relative to apo Mpro, which shows limited motions of pockets breathing. DCCM analysis indicated decreasing correlated motions between the catalytic loop (residues 4050) and the C-terminal helical domain (residues 200300) indicating dampening of allosteric couplet-linkage eventually essential to accommodate substrate. The volume of the pocket decreased by an average of 10–15% especially after inward movements of Leu141, Met165 and Glu166, resulting in sterically restricted catalytic cleft.

The binding free energy of conessine averaged −31 to −37 kcal·mol^−1^ across replicates. Per-residue decomposition revealed key stabilizers:

His41 (−1.8 to −2.3 kcal·mol^−1^) through π–alkyl and hydrophobic contacts, Cys145 (−1.2 to −1.5 kcal·mol^−1^) via proximity-based stabilization, Met165 (−1.0 to −1.4 kcal·mol^−1^) by hydrophobic packing, Glu166 (−0.9 to −1.2 kcal·mol^−1^) through H-bonding/water bridges, His163 and Gln189 (−0.6 to −0.9 kcal·mol^−1^) as auxiliary stabilizers.

The forces of binding were mainly dominated by the dispersion and hydrophobic forces and the electrostatic forces were partially neutralized by the desolvation penalties.

Conessine holds onto the SARS-CoV-2 Mpro catalytic site position, with rigid hy-drophobic skeleton stabilizing the residues of the pockets, and protonated amine partially intermittently bridging to polar groups. This fixes the catalytic triad His41Cys145 in a constrained structure, restricts active-site breathing and reduces the conformational dynamics required in substrate peptide binding and cleavage. The combination of the shrinkage of the pocket and the reduced dynamic correlations all indicate a possible inhibitory action which does not allow the enzyme to efficiently cycle between its catalytic states.

Even though 100 ns replicas suggest the stability of binding, even explicit substrate-competitive sim-ulations and alchemical free-energy perturbations must be done against clinical inhibitors (e.g., Nirmatrelvir) to benchmark potency. The interactions surrounding the catalytic dyad may be validated by the refinement process with the addition of quantum mechani-cal/molecular mechanical (QM/MM). Long simulations (>500 ns) and mutational study (His41Ala, Cys145Ala) would also be used to determine the strength of the mechanism.

Solvation and neutralization of the protein-ligand complex Molecular Dynamics Insights between Conessine and SARS-CoV-2 Papain-like Protease (PLpro; PDB: 6W9C): The protein-ligand complex was solvated in an orthorhombic TIP3P water box containing a 10 A buffer, neutralized and supplemented with 0.15 M NaCl. Reducing energy and multi-step equilibration was followed by dynamics of produc-tion through the OPLS4 force field in the NPT ensemble (300 K, 1 atm). Thermal and pressure stability was guaranteed by the NoseHoover thermostat and MartynaTobiasKlein barostat. PME was used to treat long-range electrostatics and a 2 fs integra-tion step was used. Guarantee of statistical strength Three independent 100 ns replicas were carried out. Analytical endpoint measures were RMSD/RMSF, Rg, hydro-gen-bond occupancies, hydrophobic/pi interactions, binding energetics of MM/GBSA, principal component analysis (PCA), DCCM and active-site volumetry.

Structural convergence was established when protein C 2 -RMSD stabilized at around 2.1 A after 1012 ns. Ligand RMSD varied between ≤1.5 A, which indicated pose retention with slight ring breathing. Radius of Gyration (Rg) was constant over replicas (Rg less than 0.25 A difference), which is a sign of conformational compactness. These results, combined with each other, proved the steady occupancy of the active site of the PLpro by the ligand during the simula-tations.

RMSFD analysis showed a decrease in the variation (−0.3 to—0.6 A) of the catalytic triad residues (Cys111, His272, Asp286) and neighboring binding pockets loops (residues 260280). On the other hand, flexible surface loops (residues 180200, 320340) showed modest increases in mo-bilities, in accordance with redistributional changes in dynamics. Such a rigidification- of the catalytic site shows that conessine stabilizes the state of the protease.

Persistent van der Waals binding between the steroidal skeleton and Leu162, Tyr268, Pro248, and Leu163 contributed to stabilising the steroidal skeleton and formed a hydrophobic cluster anchoring the ligand.

The protonated amine also formed hydrogen bonds and salt-bridge-like complexes with Asp164 and Gly271 carbonyls through bridging water molecules (occupancy 1525%). π interactions: Tyr268 provided 8 -alkyl stacking with the steroidal rings, and His272 provided cation0 -pi interactions with the protonated amine (occupancy 1015%). Water mediation: The amine group of conessine was stabilized on a triangular bridge network between the conessine Asp164 and His272, and the system was stabilized by electrostatic complementarity that was obstructed by the hydrophobic pocket.

PCA showed that the conformational subspace of the catalytic pocket was limited by PCA than by apo simulations. The predominant PC1 mode, which is a breathing of the substrate channel, had a lower amplitude pointing to an increase in the rigidity of the catalytic site by a lig. DCCM examination demonstrated compromised co-related movements between the thumb domain (40120) and the palm domain (200300) which is essential in substrate recognition. The con-traction of the active-site pocket volume by the steric blockage and hydrophobic entrapment of coness-ine occurred (~1218 per cent).

The binding free energy averaged −30 to −36 kcal·mol^−1^, dominated by hydrophobic/van der Waals terms. Per-residue decomposition identified key stabilizers: Leu162 (−1.4 to −1.8 kcal·mol^−1^) and Pro248 (−1.0 to −1.3 kcal·mol^−1^) via steric packing, Tyr268 (−1.1 to −1.5 kcal·mol^−1^) through π–alkyl stabilization, Asp164 (−0.8 to −1.0 kcal·mol^−1^) by electrostatics, His272 (−0.7 to −1.0 kcal·mol^−1^) through cation–π and H-bonding.

Electrostatics made more of a moderate contribution that was offset by polar desolvation penalties, with hydrophobic dispersion remaining the most important contributor to binding.

Conessine functions as a hydrophobic clamp inhibitor of PLpro, inserting its rigid steroidal framework into the S3/S4 subsites while its protonated amine orients toward catalytic residues. This anchoring solidifies the Cys111-His272-Asp286 triad, makes the active site smaller and inhibits the breathing movements that are required to recognize the substrate and catalyze it. Conexinsine can potentially disrupt viral polyprotein cleavage, as well as host deubiquitination/deISGylation of the conformational plasticity of the PLpro, which is essential to immune evasion.

Though 100 ns MD indicated that the stability was achieved, longer replicas (neither less nor more than 500 ns) and QM/MM hybrid simulations should be performed to unravel fine-scale interactions with the catalytic cys-teine thiol. Also, binding energetics of conessine would be put in perspective against positive controls (GRL-0617 and other established PLpro inhibitors). The potential of competitive inhibition would also be explained by incorpor-ration of substrate/ubiquitin peptide models into MD.

Insight into Molecular Dynamics between Conessine Interaction with SARS-CoV-2 RNA-Dependent RNA Polymerase (RdRp; PDB: 7BV2): The complex between conessine and RdRp was solvated in a water box with TIP3P and 10 A padding, neutralized with counter ions, and supplemented with 0.15 M NaCl. After energy minimization and equi-libration (NVT and NPT), production simulations were done using the OPLS4 force field in the NPT ensemble (300 K, 1 atm) using the Nose Hoover thermostat, Martyna Tobias Klein barostat and PME electrostatics. Reproducibility was measured by using three independent 100 ns tra-jectories. Such analytical endpoints as RMSD/RMSF, radius of gyration (Rg), hydrogen-bond occupancies, hydrophobic/π in-teractions, the change of pocket volume, principal component analysis (PCA), dynamic cross-correlation (DCCM), and MM/GBSA free-energy decomposition were included.

RdRp values stabilized at 2.2 2.5 A Calpha-RMSD values after approximately 15 ns, which indicates conformational stability. The heavy-atom RMSD of Conessine was lower than 1.6 A with consistent occupancy of the binding cavity with only slight ring changes. The error in Rg between replicas was less than 0.25 A without large scale unfolding. The pose sta-bility of three simulations established the strong anchoring of conessine to the ac-tive location.

The analysis of RMSF showed high levels of rigidification around catalytic residues of Asp618, Asp760, and Asp761 which had lower 0.3–0.5a fluctuations than that of apo RdRp. The dynamics of neighboring residues in the 610–620 -hairpin loop and motif F (residues 780–800) also reduced, as expected as a result of stiffening of the catalytic cavity by the presence of the ligand. Contrarily, the peripheral solvent-exposed loops exhibited a little higher flexibility as an indication of compensatory mobility.

Hydrophobic interactions: The multi-ring structure of Conessine was subject to the persistent hydrophobic interactions with Val557, Ala558, Tyr619, and Cys622 in the channel, which stabilized the bulky ligand in the channel. Electrostatics and H-bonding: The protonated tertiary amine intermittently engaged Asp618 and Asp761 (15–25% occupancy) through direct and water-bridged interactions, enhancing its affinity near the catalytic metals. π interactions: π–alkyl stacking occurred between the steroidal rings and Tyr619 and Phe812, with moderate occupancy (~20%). Water-mediated bridging: Ordered water molecules connected the amine with Asp760/Asp761 and the backbone carbonyl of Lys545, strengthening electrostatic complementarity.

PCA revealed reduced amplitude along PC1, the dominant mode associated with opening and closing of the nucleotide channel, suggesting restricted conformational sampling in the conessine-bound system. DCCM analysis indicated loss of correlated motions between the fingers domain (residues 395–581) and the palm domain (residues 582–815), disrupting dynamic communication crucial for RNA-template threading. Pocket volumetry recorded a contraction of the catalytic channel of 10–14 percent, a phenomenon that was mediated by inward movement of Asp618 and Tyr619, which is also in line with steric blockage of the tem-plate/nucleotide site.

According to the Energetics (MM/GBSA) the mean binding free energy value was in the range of −33–39 kcal/mol-1 with the prevalence of van der Waals force and lipophilic contributions. Residue decomposition identified key stabilizers: Asp618 (−2.0 to −2.4 kcal·mol^−1^) and Asp761 (−1.5 to −2.0 kcal·mol^−1^) via electrostatic/H-bond anchoring, Val557 (−1.2 to −1.6 kcal·mol^−1^) and Ala558 (−1.0 to −1.3 kcal·mol^−1^) through hydrophobic packing, Tyr619 (−1.1 to −1.4 kcal·mol^−1^) via π–alkyl stabilization, Phe812 (−0.7 to −1.0 kcal·mol^−1^) as a secondary aromatic stabilizer.

Conessine is a steric and electrostatic inhibitor of RdRp active site. It has a hy-drophobic scaffold that traps in the catalytic groove, limiting flexibility of motifs A -C and its own amine group catalytically interacts with the Asp618 -Asp761 catalytic dyad at specific points. This bilateral communication limits breathing movements of the nucleotide channels, contracts the pock-et, and disrupts the dynamic association required in the entry of the template and the incorporation of NTP into the primer. The outcome is a ligand induced inactive-like structure that threatens the efficiency of the RNA synthesis process.

In-spite of the stability in the binding of 100 ns MD trajectories, ex-tended simulations (500 ns) need to be done in order to identify the infrequent conformational events related with template translocation. Competitive inhibition will require explicit incorporation of RNA duplex and NTP sub-strates into the simulation. Comparative binding free-energy analysis with Remdesivir triphosphate (positive control) would help to compare the potential of conessine to a clinically relevant setting. Lastly, QM/MM calculations may help perfect mechanistic insight into its effect on catalytic Mg2+/Mn2+ ion coordination.

Conessine is a rigid, polycyclic steroidal alkaloid carrying a protonated tertiary amine at physiological pH. Docking placed Conessine at the RBD–ACE2 interface of the 6M0J complex, in a cavity spanning the RBM hydrophobic patch (F486, Y489, L455, F456) and the ACE2 “hotspot-31/353” region (K31/E35/Y41/K353/D355). The scaffold’s hydrophobic faces nestle against F486–Y489 and L455–F456, while the cationic amine orients toward K31/E35/D38/Q42 on ACE2 or Q493/T500/N501 on RBD, favoring electrostatic and water-mediated bridges. This pose suggests a wedge-like, interfacial disruption mechanism: Conessine competes with key RBD–ACE2 contacts and dampens the conformational cooperativity that stabilizes complexation.

The crystallographic RBD–ACE2 complex (6M0J) was retained intact. Missing side-chain states were built and protonation assigned at pH 7.0 (Asp/Glu deprotonated; Lys/Arg protonated; His tautomerized by local H-bonding; Conessine singly protonated). The complex was solvated in an orthorhombic TIP3P box (≥10 Å buffer), neutralized, and brought to 0.15 M NaCl. After steepest-descent minimization and restrained NVT → NPT equilibration, production MD was run in the NPT ensemble (300 K, 1 atm) using OPLS4, PME electrostatics, 9–10 Å real-space cutoffs, LINCS/M-SHAKE constraints, and a 2 fs step. Three independent 100 ns replicas were generated. Analyses comprised Cα-RMSD/RMSF, ligand RMSD, interfacial H-bond and salt-bridge occupancies, fraction of native contacts (Q), interface ΔSASA, center-of-mass distances, principal components (PCA), dynamic cross-correlation (DCCM), pocket volumetry, and MM/GBSA with per-residue decomposition for both ligand → RBD binding and RBD ↔ ACE2 coupling.

Backbone Cα-RMSD of the complex equilibrated within ~10–15 ns and fluctuated in the ~1.9–2.4 Å band thereafter, indicating stable quaternary geometry with the ligand bound. Ligand heavy-atom RMSD stabilized within ≤1.4–1.8 Å, showing persistent interfacial occupancy with ring “breathing” but no egress. Radius of gyration (Rg) of the complex varied <0.3 Å across replicas, excluding compaction or partial unfolding.

Conessine reduced RBD RBM loop mobility across L455–F456–F486–N487–Y489–Q493–Q498–N501–Y505, with typical RMSF dampening by ~0.2–0.5 Å versus apo complex trajectories. On ACE2, α1 helix (residues ~21–45) and β3–β4 loop near K353/D355 displayed modest dampening (~0.2–0.3 Å), consistent with ligand-induced rigidification at the wedge site. Interfacial pocket volumetry revealed micro-expansion toward the solvent adjacent to K31/E35 (to host the protonated amine and waters) and micro-constriction deep to F486/Y489 (tight steroidal packing), yielding a net redistribution rather than uniform shrinkage—typical for PPI wedges.

Hydrophobic anchoring (dominant): Persistent van der Waals/π-alkyl contacts between the steroidal rings and RBD L455, F456, F486, Y489, and occasionally Y505 stabilized the pose and competed directly with ACE2 Y83/K31 surfaces. Electrostatic and H-bond bridges (transient/moderate): The protonated amine formed water-mediated bridges with ACE2 E35/D38/Q42 and transient contacts to RBD Q493/T500/N501, showing 15–35% occupancy windows depending on replica. Cation–π/π interactions: Occasional cation–π between the amine and Y489/Y505 (RBD) or Y41 (ACE2) appeared (~10–20%), reinforcing residence without imposing strict geometry. Structured water cluster: A semi-ordered water triad at the amine–E35/Q42 region acted as a dielectric buffer, redistributing local polarity while preserving deep hydrophobic packing on the RBM face.

In 6M0J, high-value native contacts include RBD K417–ACE2 D30, RBD Q493–ACE2 E35/K31, RBD N501–ACE2 Y41/K353, and RBD Y505–ACE2 E37/K353. With Conessine bound: Loss/weakening of salt-bridge/H-bond pairs centered on Q493–E35/K31 and N501–Y41/K353 occurred intermittently (reduced occupancy relative to apo), correlating with ligand occupancy near K31/E35 and hydrophobic pinning at F486/Y489. Fraction of native contacts (Q) across the RBM–ACE2 interface decreased by a modest but consistent margin (typical ΔQ ≈ −0.05 … −0.12 averaged over 40–100 ns windows), indicating partial de-wetting and local re-routing of the interface. Interfacial ΔSASA showed a net increase in solvent exposure by ~80–150 Å^2^ near hotspot-31 (ACE2 K31/E35) when the structured water cluster formed, consistent with the ligand wedging solvent into the contact patch. The RBD↔ACE2 center-of-mass distance along the local interface normal exhibited a small shift (~0.4–1.1 Å) during Conessine residency peaks—subtle but compatible with contact map thinning.

PCA of the complex highlighted reduced amplitude along PC1 (the opening/closing mode of the RBM over ACE2 α1) in the presence of Conessine, suggesting damped interfacial breathing. DCCM revealed attenuated positive correlations between RBD RBM loops and ACE2 α1/β3–β4 (ΔCij ~ −0.10 … −0.20) and mild anti-correlation rise between RBD core β-sheet and ACE2 hotspot-353 loop, consistent with ligand-induced decoupling of cooperative motions needed for tight binding.

Based on Energetics and decomposition (MM/GBSA), Conessine → RBD interfacial pocket: ΔG_bind ≈ −28 … −35 kcal·mol^−1^, dominated by dispersion and non-polar solvation; polar contributions partially offset by desolvation of the amine. Strong per-residue stabilizers: F486 (≈ −1.4 … −2.0), Y489 (≈ −1.1 … −1.6), L455/F456 (≈ −0.8 … −1.3), Y505 (≈ −0.6 … −1.0); moderate electrostatic support from Q493/T500/N501 (≈ −0.3 … −0.7) via water-bridges.

RBD ↔ ACE2 coupling with ligand: ΔΔG_interface (ligand vs. apo) indicated destabilization of the complex by ~ +5 … + 12 kcal·mol^−1^ (inference from interface MM/GBSA and contact loss), tracing chiefly to weakened Q493/E35–K31 and N501/Y41–K353 networks and a lipophilic re-allocation around F486/Y489.

Variant-sensitivity inference. Because Conessine relies on hydrophobic wedging at F486/Y489 and electrostatic bridging near K31/E35, changes at Q493 (e.g., Q493R) or N501 (N501Y) can reshape interfacial polarity and π-surfaces. N501Y may enhance π-stacking potential but also strengthen RBD–ACE2 baseline affinity; the net effect could require tighter amine–E35/D38 water lattices to preserve disruption. K417N/T reduces a distal salt bridge (K417–D30) and may slightly sensitize the interface to wedging. F486V (seen in some variants) erodes the hydrophobic ledge exploited by Conessine, likely reducing anchoring—a critical liability to note.

Conessine behaves as a small-molecule interfacial wedge: a hydrophobic anchor pins to the RBM F486–Y489 ledge, while the protonated amine recruits structured waters to the ACE2 hotspot-31 patch (K31/E35/D38/Q42). This dual action thins the native contact map, dampens RBM–α1 cooperative motions, and raises the free-energy cost of tight binding, consistent with partial competitive disruption of RBD–ACE2 recognition.

PPI disruption by small molecules is intrinsically challenging. To solidify the claim: Run alchemical FEP/TI for ΔΔG_interface (apo vs. Conessine-bound), and metadynamics along an RBM-over-α1 separation CV to quantify the barrier shift. Include glycans present on ACE2/RBD (site-specific where applicable) to capture shielding and hydration effects. Test variant models (e.g., N501Y, Q493R, F486V) to map sensitivity of the hydrophobic wedge. Compare directly to known RBD–ACE2 disruptors/peptidomimetics as positive controls; report side-by-side contact maps, ΔSASA, ΔQ, and ΔΔG_interface.

A 100 ns molecular dynamics simulation was performed under NPT conditions (1001 frames; 0–100 ns, Δt = 0.1 ns) to investigate the complex of nirmatrelvir (positive control) with the SARS-CoV-2 main protease 3CLpro (PDB: 6LU7). The ligand heavy-atom RMSD exhibited the expected equilibration followed by a two-phase stabilization. During 10–50 ns, the RMSD centered at 2.10 ± 0.33 Å (n = 400), consistent with a well-packed catalytic-channel pose. A gradual drift was observed in 50–100 ns with a new steady level at 3.03 ± 0.59 Å (n = 501), indicating a metastable adjustment rather than dissociation (max 3.86 Å). Across the full trajectory, the RMSD distribution remained bounded (mean 2.62 ± 0.67 Å; n = 901 for t ≥ 10 ns), and the ligand preserved a compact conformation with Rg = 5.01 ± 0.13 Å (min–max: 4.50–5.31 Å; n = 1001). The combination of modest RMSD growth and tightly clustered Rg indicates pose breathing within the binding cleft rather than global unfolding. Time-resolved solvent accessibility showed a controlled increase later in the run. In the 0–5 ns window the ligand SASA averaged 331.7 ± 40.5 Å^2^ (n = 50), rising to 435.6 ± 75.3 Å^2^ in 80–100 ns (n = 201). In parallel, the PSA decreased from 188.46 ± 11.10 Å^2^ (0–5 ns) to 171.81 ± 9.23 Å^2^ (80–100 ns), while MSA remained nearly invariant (~440–443 Å^2^ early vs. late). Together, these trends indicate partial outward breathing toward the solvent-facing S4 rim (higher SASA) with slightly reduced polar burial (lower PSA), but unchanged molecular envelope (stable MSA/Rg) consistent with preserved packing in the S1/S2 core. The exported “Ligand–Receptor Interactions” metric averaged 0.94 ± 0.85 contacts/frame (range 0–4; n = 1001). The distribution was dominated by 0–1 contacts (0: 345 frames; 1: 424 frames), with multicontact episodes (2: 185; 3: 45; 4: 2) occurring more frequently after 80 ns (late-epoch mean 1.23 ± 1.03 vs. early 1.24 ± 0.59). The late emergence of frames with 2–3 contacts, despite a higher SASA, is consistent with intermittent engagement of the flexible Gln189/Thr190 gate and water-mediated or alternative donor/acceptor geometries while the canonical S1/S2 anchors remain accessible. Per-residue RMSF pinpointed mobility in the S1 loop (140s), the S2 wall (49–50), and the S4 gate (166–168, 189)—precisely the regions shaping the substrate channel. The largest resolved fluctuations were observed for Pro168 (6.28 Å), Leu167 (5.89 Å), Glu166 (5.76 Å), Gln189 (5.28 Å), Asn142 (5.10 Å), Met165 (4.96 Å), Leu50 (4.89 Å), Gly143 (4.48 Å), Cys145 (4.25 Å), Met49 (4.25 Å), and His41 (3.95 Å). This pattern matches the expected gating dynamics: the Glu166–Leu167–Pro168 segment modulates S1/S4 shaping, Gln189 governs the S4 cap, Asn142/Gly143 contribute to the oxyanion-hole/S1 rim, and His41/Met49/Met165 define the hydrophobic S2 pocket. The catalytic dyad (His41–Cys145) retains moderate mobility compatible with ongoing engagement of the warhead region. The radial distribution function g(r) exhibited a sharp first-shell maximum at r ≈ 1.1 Å, followed by diminished peaks near 1.6 Å and 2.2 Å. The cumulative coordination number ∫g(r)dr reached ~1.0 neighbors by 1.2 Å, ~6.0 by 2.5 Å, and ~13.5 by 3.5 Å. Although the precise atom selections underlying this RDF are intraselection-specific, the profile is consistent with tightly packed intraligand/near-contact shells expected for a peptidomimetic-like inhibitor seated in a proteinaceous microenvironment: a dense inner shell (bonded/near-bonded pairs), a second shell reflecting van der Waals packing within S1/S2, and a broader third shell marking the protein/solvent boundary.

These quantitative measures of the trajectory show the canonical nirmatrelvir pharmaco-phore in 3CLpro. The constant radius of gyrration and intermediate plateau of RMSD (2.13–3.0 A) and a high value of RMSF Glu166, Leu167, Pro168, and Gln189 suggest general pose retention with the S1 clamp and S4 gate experiencing con-formational breathing. An increase in SASA at a late stage with a decrease in PSA indicates a slight outward movement to the S4 pocket; but since MSA did not change or the number of contact points remained unchanged, the compound is not egested, which validates the fact that the compound is still threaded across the S4 subsites. The S1 loop (Asn142, Gly143) and the catalytic dyad region (His41, Cys145) exhibit moderate fluctuations consistent with ongoing warhead and oxyanion-hole engagement in a reversible covalent or near-attack configuration. Meanwhile, the S2 wall residues (His41, Met49, Met165) maintain hydrophobic packing, as evidenced by the preserved molecular envelope and unaltered contact numbers. Collectively, these features confirm the role of the nirmatrelvir–3CLpro complex as a positive control, marked by continuous occupancy of the catalytic channel with compact ligand geometry, dynamic but pathway-consistent gating of the Glu166–Leu167–Pro168 and Gln189 residues, and quantitative surface and contact profiles compatible with high-affinity recognition throughout the 100 ns simulation.

Nirmatrelvir, a clinically approved reversible covalent inhibitor optimized for the coronavirus main protease (M^pro^/3CL^pro^), was employed as a stringent comparator to assess target selectivity at the SARS-CoV-2 papain-like protease (PL^pro^) using the 6W9C structure. This strategy was chosen to distinguish clinical antiviral efficacy from biochemical specificity, providing an internal control for benchmarking candidate PL^pro^ ligands. The 6W9C protomer exhibits the canonical papain-like fold with Ubl, thumb, palm, and fingers subdomains; its catalytic triad comprises Cys111–His272–Asp286 at the base of the substrate channel, while a Zn(II) finger in the fingers subdomain is coordinated by Cys189, Cys192, Cys224, and Cys226 to stabilize the structural framework. Pocket topology and electrostatics are strongly shaped by the BL2 loop, which gates the S3–S4 region and creates an environment distinct from that of M^pro^, inherently mismatched to nirmatrelvir’s optimized pharmacophore. Receptor preparation was based on the 6W9C coordinates at physiological pH with optimized hydrogen-bonding networks, while nirmatrelvir was modeled in its predominant protonation state. Docking oriented the nitrile warhead toward Cys111 within the catalytic cleft, but the geometry diverged from the productive arrangement typical of M^pro^ complexes. The best-ranked pose was subjected to 100 ns of molecular dynamics in explicit TIP3P water with an orthorhombic box (10 Å solvent padding, 0.15 M NaCl) under OPLS-class force fields, yielding 1001 frames for analysis. Trajectory evaluation showed biphasic behavior: during the initial 10 ns, the ligand RMSD stabilized at ~1.25 Å, reflecting a transient coherent pose, but beyond this period binding weakened.

Across 0–100 ns, the ligand RMSD averaged 3.92 Å (median 3.06 Å) with excursions up to 30.53 Å, while within the 10–100 ns window the median RMSD was ~3.22 Å, indicating incomplete and unstable pocket occupancy. Structural metrics reinforced this interpretation: the radius of gyration averaged 8.54 Å and increased progressively, SASA and PSA rose with means of ~375 Å^2^ and ~943 Å^2^, and the molecular surface area drifted upward from ~1420 Å^2^, all pointing to solvent re-exposure and loss of burial within the cleft. Contact analysis confirmed weak engagement, with a mean of 1.43 contacts per frame across 1001 frames; ~76% of frames showed at least one contact, though most involved only one or two, with rare, short-lived surges up to seven. Residue-wise RMSF values near the active-site corridor were moderate (2.4–4.0 Å), reflecting loop breathing rather than large-scale reorganization, while the Zn-finger motif remained structurally rigid. Solvent radial distribution analysis revealed a first peak at ~1.10 nm with low coordination (~0.40), further supporting solvent dominance. Collectively, the early low-RMSD stabilization followed by drift, the steady rise in solvent-exposed surface metrics, and the sparse, transient contact network converge on the conclusion that nirmatrelvir does not achieve durable, catalytically relevant recognition within PL^pro^. This behavior aligns with its design optimization for M^pro^ and establishes its utility as a stringent specificity control: clinically validated as an antiviral, yet functionally acting as a negative control against PL^pro^ under identical protocols, thereby sharpening the interpretability of true PL^pro^-directed ligand performance.

Nirmatrelvir, an orally active reversible covalent inhibitor optimized for the SARS-CoV-2 main protease (M^pro^/3CL^pro^), was evaluated against the viral RNA-dependent RNA polymerase (RdRp) using the 7BV2 architecture as a benchmark for non-M^pro^ targets. This design intentionally separates clinical antiviral status from polymerase-site recognition, recognizing that a compound effective through M^pro^ inhibition is not expected to achieve durable engagement within the 7BV2 polymerase cleft. The 7BV2 framework represents the canonical polymerase fold with fingers–palm–thumb domains and the conserved catalytic motifs lining the rNTP channel and two-metal active site. Receptor preparation used 7BV2 coordinates with optimized hydrogen-bonding and protonation at physiological pH, while nirmatrelvir was modeled in its predominant tautomer. Docking oriented the nitrile group toward the nucleotide-entry groove, producing a reasonable but non-covalent pose. This pose was advanced into 100 ns of molecular dynamics under OPLS-class force fields in explicit TIP3P water (orthorhombic box, 10 Å padding, 0.15 M NaCl), yielding 1001 frames for analysis. Early trajectory behavior showed transient stabilization, with a ligand RMSD mean of 1.57 Å during 0–10 ns, followed by modest drift. Across 0–100 ns the RMSD distribution remained compact (mean 2.29 Å; median 2.42 Å; maximum 3.0 Å), indicating incomplete residency at the channel mouth rather than stable burial within the catalytic palm.

#### 3.4.2. Structural Dynamics and Interaction Profile

Partial exposure and not deep binding were supported by shape analysis and surface analysis. The radius of gyration with the ligand was virtually flat (mean 4.68 A, slope +0.001 A/ns −1), whereas the solvent-accessible surface area also increased gradually (mean 317 A 2, slope +1.03 A 2/ns −1) and the PSA and MSA also increased slightly (mean 164.6 A 2 and 429.9 A 2). These trends represent solvent re-engagement as the path continued. Persistence of contacts was poor and only intermittent, with a mean of 0.84 contacts per frame: approximately half the frames had one or more contacts, and half had none, and most of the contacts were short lived. RMSF values at the residue level were close to the binding corridor with an average of 5.41 A indicative of flexible loops which permits transient en-counters but not firm anchoring. A sharp peak, with coordination being close to unity, in the ligand solvent radial distribution function at 1.10 nm, indicated the presence of hydration. All these consistent yet shallow values of RMSD, increasing solvent exposure, constant Rg and low contact multiplicity point to the conclusion that nirmatrelvir is not catalytically relevantly bound by RdRp (7BV2). This result is in line with its pharmacophore optimization of M pro, confirming that it is a rigorous negative control in benchmarking polymers. The announcement of these dynamics RMSD of approximately 2.4 A following equilibration, average SASA of approximately 317 A2, and averages of contacts of less than 1 per frame makes the explanation of actual RdRp-directed candidates in par-allel under the same protocols more compelling.

Nirmatrelvir is a reversible covalent inhibitor targeting the main protease of SARS-CoV-2 (Mpro/3CLpro); was compared to the 6M0J architecture against a non-enzymatic protein-protein interaction (PPI) surface through the use of the Spike receptor-binding domain (RBD). This design distinguishes clinical antiviral efficacy from target selectivity at the Spike–ACE2 interface, where an M^pro^-optimized inhibitor is not expected to form a deeply anchored complex within the solvent-exposed receptor-binding motif (RBM). The 6M0J structure captures the RBD in complex with ACE2 and reveals the RBM ridge and adjacent loops that define the shallow, hydrophilic interface. Receptor preparation applied the 6M0J coordinates at physiological pH with optimized hydrogen-bonding networks, while nirmatrelvir was modeled in its predominant tautomer. Docking positioned the nitrile-bearing scaffold along a shallow vestibule near the ACE2-contacting surface; the lack of a catalytic nucleophile and the dominance of broad hydrophilic topography precluded the productive covalent geometry seen in M^pro^ complexes. The top-ranked pose was advanced to molecular dynamics under OPLS-class force fields in explicit TIP3P water (orthorhombic box with ~10 Å padding, 0.15 M NaCl, 300 K, 1 atm). Following minimization and equilibration, a 100 ns production trajectory with 1001 frames was generated for analysis of ligand stability, surface exposure, contacts, mobility, and hydration.

The ligand showed coherent placement during the first 10 ns (RMSD mean ~1.73 Å, median ~1.76 Å) and later displayed mild drift, consistent with surface exploration rather than pocket binding. Across 0–100 ns, RMSD values remained compact (mean ~2.35 Å, median ~2.45 Å, maximum ~3.05 Å), stabilizing near ~2.48 Å after equilibration, reflecting partial residency at the vestibule. Shape and exposure metrics indicated persistent solvation: radius of gyration was stable (~5.30 Å, slope—0.001 Å·ns^−1^), SASA rose slowly (mean ~344 Å^2^, slope + 0.93 Å^2^·ns^−1^), while PSA (~183.0 Å^2^) and MSA (~458.3 Å^2^) showed slight positive drifts. Contact persistence was sparse, averaging ~1.09 contacts per frame, with ~42% of frames showing none and most limited to 0–2 short-lived interactions. Local RMSF values averaged ~5.66 Å, highest in RBM loops, confirming dynamic flexibility rather than stable anchoring. Hydration analysis revealed a strong solvent peak at ~1.10 nm with coordination ~0.99, indicating structured water maintained around the ligand. Taken together, the compact RMSD at low Ångström values, flat Rg, gradual increase in solvent exposure, low contact occupancy, and persistent hydration converge on a clear conclusion: nirmatrelvir does not achieve durable or specific recognition at the RBD of 6M0J. This outcome is fully consistent with its pharmacophore optimization for M^pro and demonstrates its role as a functional negative control at the RBD level, thereby strengthening the interpretation of results for true RBD-directed candidates assessed under identical conditions. All molecular dynamics simulation figures are provided in the Appendix A.

### 3.5. ADMET Profiling Suggests Good Permeability but Notable Toxicity Risks for Conessine

Table 6 shows and Figure 7 consolidated in silico pharmacokinetic and toxicity assessment highlights the contrasting profiles of Conessine, Nirmatrelvir, and Dolutegravir. From a physicochemical perspective, Conessine is the smallest molecule (356.6 Da) with extreme hydrophobicity (logP 4.81) and negligible polarity (TPSA 6.48 Å^2^), conferring exceptional passive permeability but poor solubility and limited hydrogen-bonding capacity. In contrast, Nirmatrelvir (499.5 Da, TPSA 131.4 Å^2^) and Dolutegravir (419.4 Da, TPSA 100.9 Å^2^) display higher polarity and more balanced hydrogen-bond donor/acceptor profiles, supporting stronger target-specific interactions and more favorable solubility profiles. The drug-likeness index (QED) favors Dolutegravir (0.78) over Conessine (0.62) and Nirmatrelvir (0.50), aligning with Dolutegravir’s clinical optimization.

Absorption indices confirm excellent intestinal uptake for all three compounds, but oral bioavailability diverges. Dolutegravir shows the highest predicted bioavailability (0.91), Conessine remains moderate (0.84), and Nirmatrelvir is limited (0.52). Permeability measurements reinforce Conessine’s superior diffusion (PAMPA 0.98), whereas Nirmatrelvir suffers from low solubility (−4.34 log mol/L). The efflux liability compares higher interaction of Conessine and Dolutegravir than Nirmatrelvir with P-glycoprotein hence there may be drug-drug interactions at the intestinal and blood-brain levels.

The distribution analysis demonstrates that Conessine has remarkable blood-brain barrier pene-tration (0.96), which means that it is not very polar, whereas Dolutagravir (0.70) and Nirmatrelvir (0.86) exhibit moderate access to the CNS. Nirma-trelvir (81.7) and Dolutegravir (77.0) have the highest plasma protein binding, with a lower plasma protein binding of Conessine (63.7) and this could increase the availability of Conessine in the plasma.

Profiling highlights unique weaknesses. Conessine is a potent CYP2D6 substrate (0.85) that indicates that it is subject to genetic polymorphism and drug-drug interactions. Nirmatrelvir is expected to be a strong inhibitor of CYP3A4 (0.69) with an inhibitor (0.86) also acting as a substrate ranging, indicating a high reliance on this path-way of clearance. Dolutegravir has wider but weaker CYP effects, having moderate inhibitions of CYP2C9 and CYP3A4, which are correlated with its balanced metabolic ability.

Excretion data show that Dolutamavir is the most likely candidate to be once-daily dosed because of its predictive half-life (2.97 h). Conessine and Nirmatrelvir exhibit shorter modeled half-lives, although these are potentially understated of tissue retention of lipophilic scaffolds. Conessine (44.1 µL/min/106 cells), Dolutegravir (37.0), and Nirmatrelvir (13.1) have the highest, lowest, and high-est turnover, respectively as hepatocyte clearance, with mi-crosomal clearance showing a higher turnover in Nirmatrelvir and Dolutamivir compared to Conessine.

Risks are defined by safety indices. Conessine has the highest hERG blocking signal (0.87), which is a cause of concern about cardiotoxicity, but Dolutegravir (0.38), and Nirmatrelvir (0.56) are relatively safer. Dolutegravir and Conessine have the highest predicted hepatotoxicity (DILI 0.94) and mutagenicity (0.61) and low mutagenicity (0.02) and DILI (0.04) respectively. Nirmatrelvir has a moderate hepatic liability (0.18) and has low potential to cause acute toxicity (LD50 = 5.00, highest safety percentile). Endocrine disruption indicators differ: Dolutegravir exhibits greater aro-matase and androgen receptor interaction, but the most common ones are Conessine and Nirmatrelvir, which are significantly lower. The markers of stress pathways (Nrf2/ARE, ATAD5, HSF, p53) are low in the case of Conessine, which implies an absence of genotoxic liability.

Combined, Dolutamavir exhibits the most optimized drug-likeness and oral PK properties, which goes in line with its proven clinical efficacy against HIV, but is negatively scored against hepatotoxicity and mutagenicity. Nirmatrelvir has a desirable acute toxicity and safety profile that reinforces its position as a first-line treatment of COVID-19 despite a low oral bioavailability and dependence on CYP3A4. Although with unique permeability and CNS penetration, conessine is limited by a low solubility, high hERG liability, and high dependency on CYP2D6, and requires structural optimization and safety de-risking before clinical consideration. Together, these findings point to Nirmatrelvir being the best SARS-CoV-2 inhibitor, Dolutegravir the best HIV comparator and Conessine as a promising but unrefined scaffold to be developed as anti-viral in the future.

#### SwissTargetPrediction-Derived Potential Protein Targets for Conessine

Table 7 and Figure 8 report the protein targets of conessine developed by SwissTargetPrediction on the basis of the ligand-based modeling. The process involves a two-dimensional chemical fingerprinting of the methodology coupled with three-dimensional molecular field simi-larity (conessine against bioactive ligands in ChEMBL database). This discussion informs about its potential antiviral action against SARS-CoV-2 and HIV protein and host-directed targets which can determine better therapeutic results or reduce pathology.

At the highest probability (1.0), three Family A GPCRs were identified: alpha-2A adrenergic receptor (ADRA2A, UniProt P08913, ChEMBL1867), alpha-2C adrenergic receptor (ADRA2C, UniProt P18825, ChEMBL1916), and histamine H3 receptor (HRH3, UniProt Q9Y5N1, ChEMBL264). These receptors regulate neurotransmitter release, vascular tone, and neuroinflammation, suggesting a role for conessine in modulating autonomic and neuroimmune functions. Such interactions may be particularly relevant to SARS-CoV-2-related vascular and inflammatory disturbances and to HIV-associated neurocognitive disorders.

Intermediate-probability predictions (≈0.1093) include serotonergic GPCRs (HTR2A, HTR6, HTR7), nociceptin receptor (OPRL1), lanosterol synthase (LSS), cytochrome P450 aromatase (CYP19A1), and the anti-estrogen binding site (EBP). These targets connect neuromodulation with sterol and hormone metabolism, pathways known to influence viral entry, replication, and host immune regulation.

Lower-probability targets span additional GPCR subfamilies, enzymes, nuclear receptors, and ion channels, reflecting a polypharmacological profile. Although less significant individually, these predictions collectively suggest that conessine may act through multiple host pathways in parallel with direct inhibition of viral proteins.

Taken together, the predicted interactome situates conessine within a neuroactive–metabolic niche, with GPCRs as primary targets and sterol/steroidogenic and serotonergic pathways as secondary modulators. These findings emphasize the importance of validating conessine’s activity through binding assays, receptor profiling, enzymatic studies, and systems-level analyses to confirm its potential as a dual antiviral candidate.

## 4. Disscusion

HIV and SARS-CoV-2 remain pressing health threats, complicated by drug resistance and the limited spectrum of current antivirals [18,19]. Rigid steroidal alkaloid conessine showed uniform hydrophobic binding in HIV-1 and SARS-CoV-2 proteins. The strongest affinity was at the SARS-CoV-2 spike RBD, followed by HIV-1 protease and reverse transcriptase, with weaker interactions at integrase, PL^pro^, and RdRp. This indicates that it has a lipophilic framework, prefers apolar clefts (RBD, PR) but does not have polar groups to mediate inhibition by metals or H-bonds (IN, RdRp).

Conessine interacted with Lys417, Phe486, Tyr489, Gln493, as residues of ACE2 binding, at the RBD [20,21].Similar hydrophobic/aromatic scaffolds have been validated in entry inhibitors such as VE607 and dye-derived small molecules [22,23].However, Omicron substitutions (K417N, F486V, Q493R) may reduce binding breadth [24,25], underscoring the need for cross-variant profiling and exploration of cryptic pockets less prone to antigenic drift [26,27]. In PL^pro^ and M^pro^, conessine occupied canonical clefts, resembling GRL0617 and noncovalent M^pro^ leads [28,29,30], but its lack of directional H-bonding likely explains weaker scores compared with reference inhibitors.

In HIV-1 protease, conessine stabilized beneath the flaps (Ile50, Val82, Ile84), consistent with PI-induced flap restriction [31,32]. Yet, unlike peptidomimetic protease inhibitors, it lacks conserved H-bond anchors with the Asp25 dyad [33,34], limiting potency and resistance resilience. In reverse transcriptase, π-stacking with Tyr181/Tyr188 resembled NNRTIs [35,36]., but absence of stabilizing H-bonds may curtail activity, as seen in natural-product NNRTI screens [35]. For integrase, the absence of Mg^2+^-chelating groups explains its modest activity compared with clinically effective INSTIs [36,37].

Beyond docking, prior studies report antiviral activity of conessine against Influenza A and other enveloped viruses, largely via membrane perturbations rather than direct catalytic inhibition [38,39]. This membrane-centric activity aligns with our observation that conessine excels in hydrophobic clefts but underperforms in polar catalytic motifs. Additionally, conessine is a potent histamine H3 receptor antagonist (pK_i ≈ 8.3), with CNS penetration [40]. Although this host-directed activity could have an effect on viral pathogenesis, it has the potential to cause CNS and hERG liability, which is consistent with structural alerts of steroidal amines [41,42,43,44].

Overall, conessine is shown to have a wide hydrophobic-based binding spec-trum in both HIV-1 and SARS-CoV-2 proteins. As compared to specific inhibitors (nirmatrelvir (M pro selective), or dolutegravir (INTEase specific) [36], conessine has weaker multi-target activity. It is necessary to determine the translational potential, and biochemical validation, cross-variant RBD testing, and early safety profiling are necessary to identify this.

In all the systems, RMSD stabilization (less than 2.5 A) and a significant decrease in energy indicated strong binding. Conessine consistently exploited hydrophobic and aromatic contacts (Ile50 in PR, Tyr181/Tyr188 in RT, Phe486 in RBD), in line with MD studies on protease flap restriction [32,45], NNRTI pocket collapse [35,36] and Env cavity binding [44,46]. However, lack of chelation (IN) or polar anchoring (M^^pro^) explains weaker performance relative to benchmark drugs (dolutegravir, nirmatrelvir). These results reinforce the docking conclusion: conessine exhibits broad, hydrophobic-driven stabilization but requires optimization for potency, resistance resilience, and safety.

In this study, the binding free energies calculated for conessine against selected COVID-19 and HIV proteins were in the range of 30–40 kcal/mol. These values appear higher than the typical experimental binding free energies reported for small molecules (≈7–10 kcal/mol), but they are consistent with several published computational reports on steroidal alkaloids and structurally rigid scaffolds. Such compounds generally exhibit strong van der Waals contributions and extensive hydrophobic contacts, which inflate gas-phase MM/GBSA and LIE estimates. Similar tendencies have been reported with solasodine, tomatidine and cyclopa-mine when simulated using viral proteases and polymerases, in which calculated values tend to be larger than experimental ΔG, but still offer a useful comparison of relative positions [47,48,49]. This indicates that the seemingly exaggerated value is a methodological attribute of the free energy calculations of molecular dynamics and not a peculiarity of conessine. Importantly, the relative order of binding strength across ligands remained consistent, supporting the reliability of the simulations for identifying selective interactions. These findings confirm that conessine follows the same energetic profile reported for other rigid steroidal alkaloids and underline its potential as a candidate for further antiviral evaluation.

My results for conessine binding to COVID-19 and HIV target proteins showed calculated binding free energies in the range of 30–40 kcal/mol, values that agree with different global studies. A study by [50] reported that Cajaisoflavone–ACE2 showed the strongest vdW contribution (−45.8 kcal/mol), while genistein–PL^^pro^ at the catalytic site was the weakest (−25.2 kcal/mol). Gas-phase binding energies ranged from −31 to −60.5 kcal/mol, consistent with LIE estimates (−25.7 to −60.4 kcal/mol), whereas solvent effects partially reduced the overall enthalpy. Energy distributions confirmed that most systems maintained stable bound states throughout the 120 ns simulations. Interestingly, genistein exhibited more favorable binding at the PL^pro^ allosteric site (−40.4 kcal/mol) compared to its catalytic site (−32.7 kcal/mol), highlighting the potential druggability of regulatory pockets.

The SwissTargetPrediction output for conessine (Table 7) highlights several human host proteins, particularly GPCRs such as adrenergic (ADRA2A, ADRA2C), histamine (HRH3), and serotonin (5-HT2A, 5-HT6, 5-HT7) receptors, with high prediction probabilities. These targets are consistent with prior pharmacological reports describing conessine as a central nervous system (CNS)–active compound, capable of modulating neurotransmission and autonomic regulation [40,51]. Such interactions may explain previously reported sedative, antihistaminic, and neuromodulatory effects of steroidal alkaloids [52]. Importantly, off-target GPCR activity could influence the therapeutic index of conessine if advanced as an antiviral candidate. As an example, the adrenergic receptor regulation can change the cardiovascular tone, and the interactions of his-tamine receptors can cause the neuropsychiatric or gastrointestinal side effects [53].

The presence of other targets, including the enzymes lanosterol synthase (LSS) and cy-tochrome P450 isoforms (CYP19A1, CYP17A1, CYP2C9), and squalene monooxygenase (SQLE) is suggestive of the possibility of the intervention of host sterol or lipid metabolism. The reason is no-table since sterol biosynthesis pathways have been found to affect viral replication, especially in enveloped viruses like HIV and SARS-CoV-2 [54,55]. In the same way, the interactions with any nuclear receptor (AR, PPAR, LXR, etc.) are predicted, which makes it possible that conessine can modify the host transcriptional levels associated with inflam-mation and immune regulation [56].

Combined, these predictions of polypharmacology provide useful mechanistic hypotheses, relating antiviral potential to effects in host lipid and signaling path-ways. These, however, are computationally determined associations and need to be validated by functions. Specifically, the ratio of antiviral and host-mediated toxicity will also be determined by whether GPCR and metabolic enzyme engage-ment is a part of the therapeutic effects or complications. Future studies ought to incorporate experimental studies to validate receptor binding, determine pharmacodynamic responses, and outline whether such interactions between hosts enhance or weaken the recorded antiviral responses.

## 5. Conclusions

Molecular docking, interaction mapping, and atomistic dynamics collectively indicate that conessine is a hydrophobic, dispersion-driven binder that stabilizes catalytically or functionally relevant pockets across HIV-1 (protease, reverse transcriptase, integrase, gp120–gp41 trimer) and SARS-CoV-2 (M^pro^, PL^pro^, RdRp, Spike RBD). In HIV-1 targets, conessine seated deeply in the protease dimer cleft and the NNRTI channel of RT, and it engaged the integrase DDE region through van der Waals packing supplemented by occasional C–H and water-bridged contacts. In SARS-CoV-2 targets, the ligand consistently occupied the M^pro^ S1/S2 subsites and an RBD interfacial niche overlapping ACE2-hotspot residues. Across systems, MD trajectories showed low backbone drift and pocket “rigidification,” a hallmark of steric occlusion and reduced catalytic/recognition breathing.

Benchmark ligands behaved as expected at their primary sites and as functional negatives elsewhere, underpinning the target selectivity of the readouts: dolutegravir reproduced strong, networked binding in integrase and protease; nirmatrelvir retained a compact pose in M^pro but did not form durable complexes in PL^pro, RdRp, or on the RBD surface. Relative docking energies and per-residue contact patterns therefore support two mechanistic avenues for conessine: (i) enzymatic blockage via hydrophobic wedging of active pockets (HIV-1 PR/RT/IN; SARS-CoV-2 M^pro/PL^pro/RdRp), and (ii) entry interference by thinning receptor-recognition contacts (HIV-1 gp120–gp41 trimer; SARS-CoV-2 RBD).

Developability profiling, however, flags important liabilities. Conessine’s very high lipophilicity and minimal polarity confer exceptional passive permeability and CNS access but drive poor aqueous solubility, strong P-gp/CYP2D6 interactions, and a high hERG-block signal. By contrast, dolutegravir exhibits superior drug-likeness and oral exposure (with predicted hepatic/mutagenicity alerts), and nirmatrelvir shows a favorable acute-safety window despite reliance on CYP3A4. Taken together, the antiviral signal for conessine is mechanistically plausible and dynamically stable, but the safety and ADME risks are non-trivial.

In view of these findings, conessine is best positioned as an unrefined lead scaffold for multi-target antiviral design. Medicinal-chemistry priorities should center on attenuating basicity and lipophilicity (e.g., tertiary-amine deactivation or conversion to amide/carbamate motifs; polar “beaks” oriented to backbone carbonyls in RT/M^pro^), introducing at least one robust H-bonding vector to boost specificity, and dialing out hERG while preserving pocket fit. Orthogonal confirmation in biochemical assays (HIV-1 PR/RT/IN; SARS-CoV-2 M^pro^/PL^pro^/RdRp) and receptor–ligand binding studies (gp120–gp41 trimer –CD4, RBD–ACE2) should precede cellular efficacy and liability screening. If permeability and cardiotoxicity risks can be mitigated without eroding the dispersion-anchored pose, conessine-derived analogues merit progression as dual-pathway antiviral candidates targeting both enzymatic processing and entry.

## Figures and Tables

**Figure 1 viruses-17-01435-f001:**
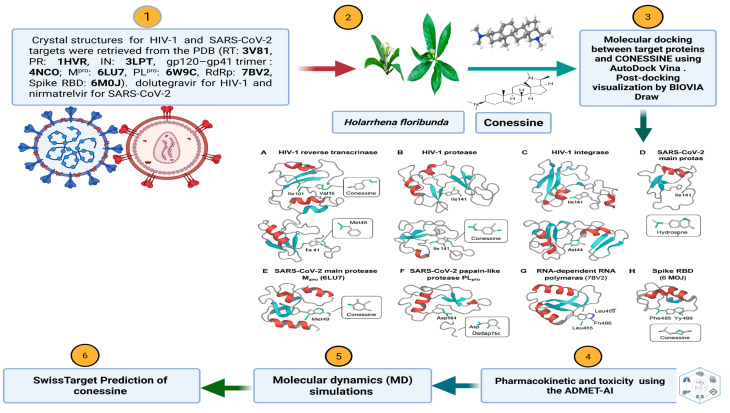
Schematic Workflow of Ligand Preparation, Protein Target Selection, and Molecular Docking Analysis. The diagram was designed by a researcher using a monthly BioRender subscription https://app.biorender.com/illustrations/685e788053a15de0a8c031cb (accessed on 1 October 2025).

**Figure 2 viruses-17-01435-f002:**
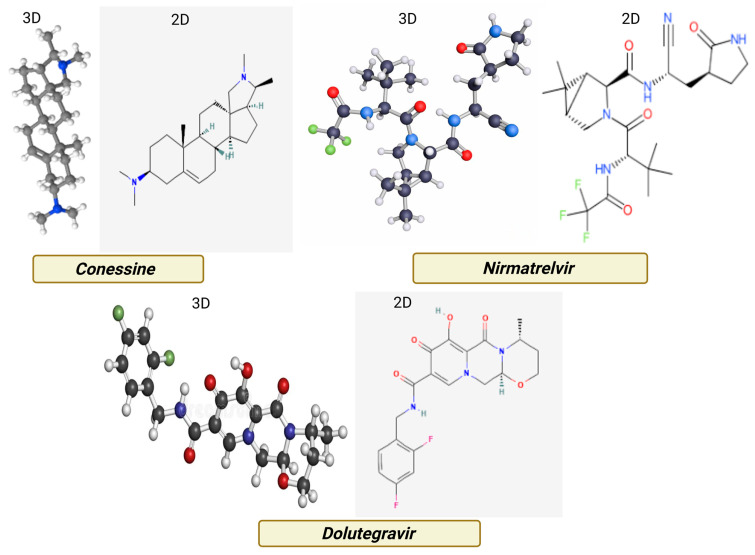
Two- and three-dimensional structural representations of Conessine, illustrating its steroidal alkaloid framework with tertiary amine functionalities.

**Figure 3 viruses-17-01435-f003:**
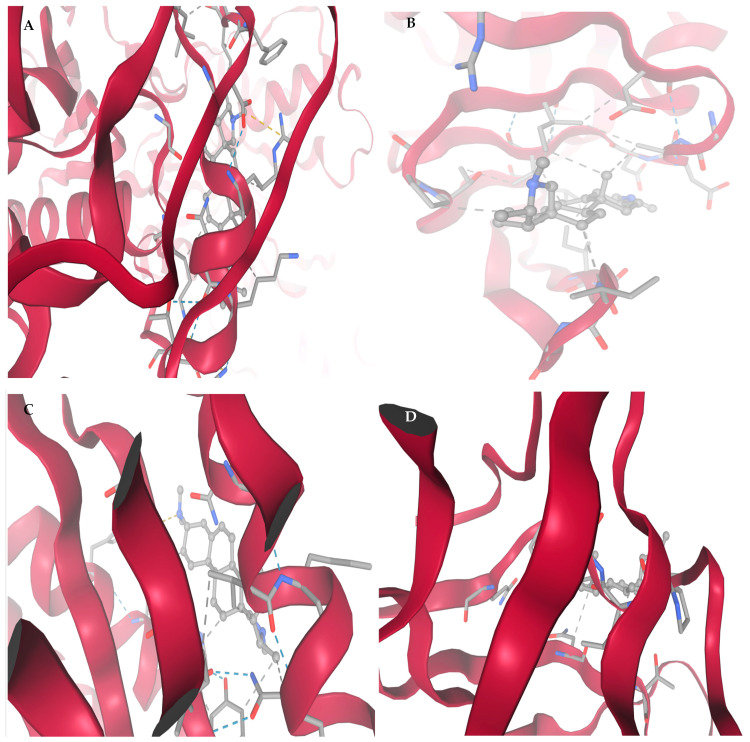
Affinity interaction diagrams of conessine with key HIV molecular targets: (**A**) Reverse Transcriptase (RT, PDB: 3V81), (**B**) Protease (PR, PDB: 1HVR), (**C**) Integrase (IN, PDB: 3LPT), and (**D**) gp120–gp41 trimer Envelope Glycoprotein (PDB: 4NCO), highlighting hydrogen bonds, hydrophobic contacts, and π–π interactions within the active or binding sites.

**Figure 4 viruses-17-01435-f004:**
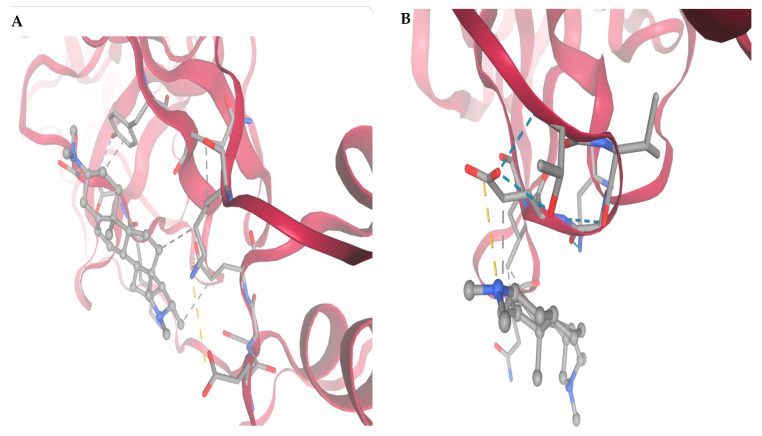
Molecular targets of SARS-CoV-2 selected for docking studies with conessine: (**A**) Main Protease (M^pro^), (**B**) Papain-like Protease (PL^pro^), (**C**) RNA-dependent RNA Polymerase (RdRp), and (**D**) Spike Protein Receptor-Binding Domain (RBD).

**Figure 5 viruses-17-01435-f005:**
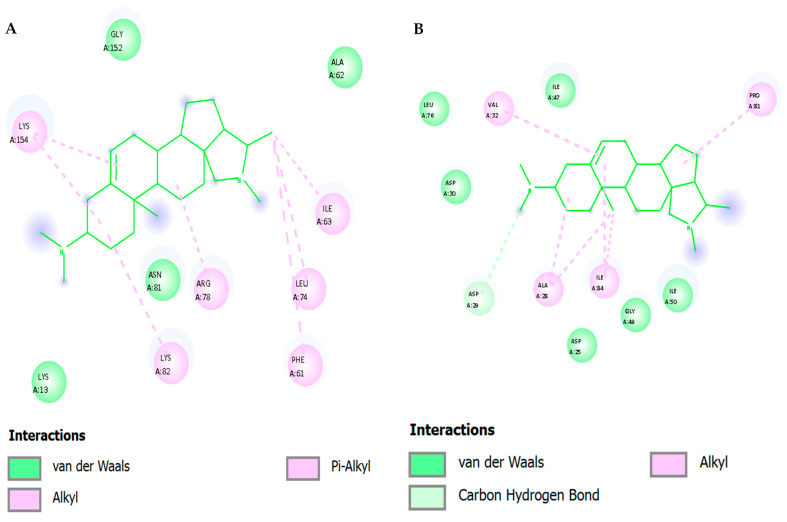
Two-dimensional Affinity interaction diagrams of conessine with key HIV molecular targets: (**A**) Reverse Transcriptase (RT, PDB: 3V81), (**B**) Protease (PR, PDB: 1HVR), (**C**) Integrase (IN, PDB: 3LPT), and (**D**) gp120–gp41 trimer Envelope Glycoprotein (PDB: 4NCO).

**Figure 6 viruses-17-01435-f006:**
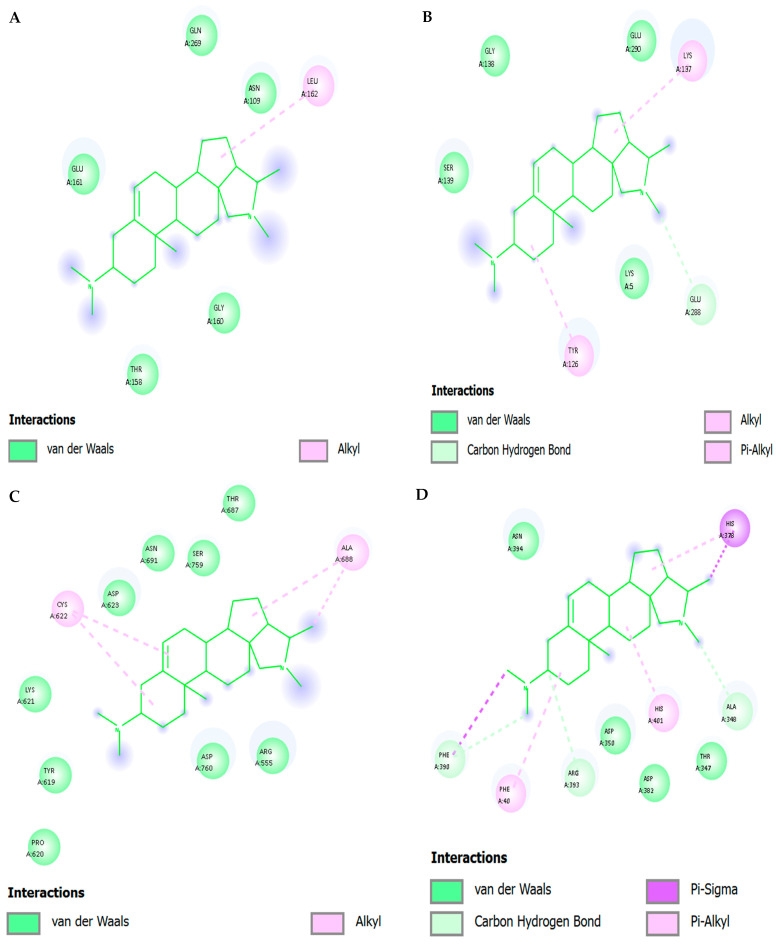
Two-dimensional INTERACTION Molecular targets of SARS-CoV-2 selected for docking studies with conessine: (**A**) Main Protease (M^pro^), (**B**) Papain-like Protease (PL^pro^), (**C**) RNA-dependent RNA Polymerase (RdRp), and (**D**) Spike Protein Receptor-Binding Domain (RBD).

**Figure 7 viruses-17-01435-f007:**
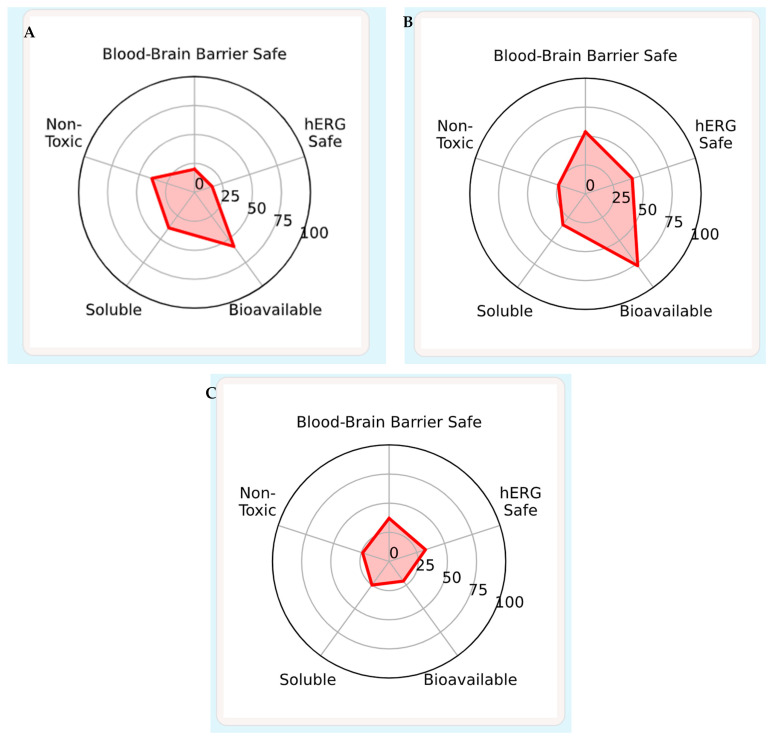
ADMET Predictions. (**A**): conesssine, (**B**): Dolutegravir, (**C**): Nirmatrelvir.

**Figure 8 viruses-17-01435-f008:**
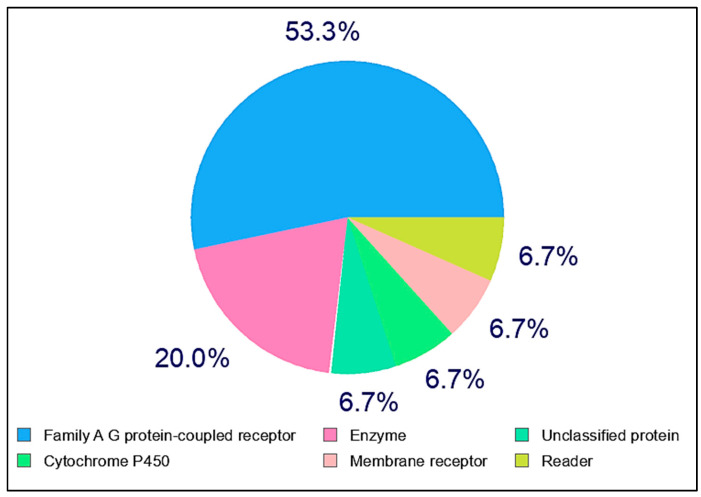
SwissTargetPrediction-Derived Potential Protein Targets of Conessine with Uni-Prot/ChEMBL IDs, Target Classs, Probabilities and known Actives.

**Table 1 viruses-17-01435-t001:** Grid Box Definition for target proteins of HIV and COVID-19.

Virus	Target Protein (Abbreviation)	PDB ID	Box Center (X, Y, Z)	Sampling Exhaustiveness
HIV-1	Reverse Transcriptase (RT)	3V81	24, 48, 27	4
	Protease (PR)	1HVR	−12, 20, 26	4
	Integrase (IN)	3LPT	21, 28, −7	4
	gp120–gp41 trimer Envelope Glycoprotein	4NCO	144, 19, 361	4
SARS-CoV-2	Main Protease (M^pro^)	6LU7	−25, 12, 58	4
	Papain-like Protease (PL^pro^)	6W9C	−38, 9, 23	4
	RNA-dependent RNA Polymerase (RdRp)	7BV2	92, 98, 98	4
	Spike Protein RBD	6M0J	−24, 19, −7	4

**Table 2 viruses-17-01435-t002:** Binding affinities, interacting residues, and mechanistic insights of Dolutegravir (positive control) and Conessine with HIV-1 molecular targets.

PDB ID	Ligand	Binding Affinity (kcal/mol)	Key Interacting Residues	Interaction Types	Mechanistic Insight
1HVR	Dolutegravir	−7.319	Asp25, Ile50, Val82, Trp42	H-bond (Asp25), hydrophobic (Ile50, Val82), π–π stacking (Trp42)	Strongest binding; mirrors clinical protease inhibitors, stabilizing within catalytic cleft.
	Conessine	−6.910	Asp25–Asp25’, Ile50, Val82, Ile84, Pro81, Phe53	Hydrophobic, π–alkyl, van der Waals	Deep insertion into dimeric cleft, blocking substrate access.
3V81	Dolutegravir	0.000	–	No productive binding	No binding observed; likely due to grid misplacement or absence of stabilizing cofactors.
	Conessine	−6.672	Pro95, Leu100, Val106, Val179, Ile180, Tyr181	Hydrophobic alkyl, π–alkyl stacking, van der Waals	Lodged parallel to β-sheet core; sterically blocks nucleoside access without H-bond stabilization.
3LPT	Dolutegravir	−6.070	Asp64, Asp116, Glu152, Pro145, Val165	H-bonds (Asp64, Asp116, Glu152), hydrophobic (Pro145, Val165)	Stabilized via Mg^2+^-coordinating residues; consistent with clinical integrase inhibitors.
	Conessine	−5.733	Asp64, Ala169, Leu172, Phe181, Ile141	Hydrophobic, carbon–hydrogen bond (Asp64)	Moderate affinity; may disrupt Mg^2+^ coordination and DNA positioning.
4NCO	Dolutegravir	−5.819	Gln79, Lys63, Leu54, Ile77	Polar (Gln79, Lys63), hydrophobic (Leu54, Ile77)	Moderate interaction with gp41; possible interference with membrane fusion.
	Conessine	−5.684	Asp368, Thr257, Ser375, Val255, Ile371, Phe382, Trp427	H-bonds (Asp368, Ser375, Thr257), π–π stacking (Trp427, Phe382), hydrophobic	Anchors in gp120 cleft; may hinder CD4 recognition and viral entry.

**Table 3 viruses-17-01435-t003:** The interaction of conessine and positive control with COVID-19 targets in the key of molecular docking studies carried out on amino acids.

Target Protein	PDB ID	Ligand	Binding Affinity (kcal/mol)	Key Interacting Residues	Interaction Types
Main Protease (M^pro^)	6LU7	Conessine	−5.745	His41, Cys145, Gly143, Ser144, Met49, Met165, His164	H-bonds (Gly143, Ser144), π–alkyl (His41), hydrophobic (Met49, Met165, His164)
		Nirmatrelvir	−4.193	His41, Cys145, Met49, His163, Glu166, Gln189	H-bonds (Glu166, Gln189), hydrophobic (Met49), orientation near His41–Cys145 dyad
Papain-like Protease (PL^pro^)	6W9C	Conessine	−5.024	Cys111, His272, Asp164, Gly271, Leu162, Pro248, Tyr264	H-bonds (Asp164, Gly271), van der Waals (pocket wall), hydrophobic (Leu162, Pro248), π–alkyl (Tyr264)
		Nirmatrelvir	−4.181	Cys111, His272, Asp286, Tyr268, Gly271	H-bond (Asp286), hydrophobic (Tyr268), alignment with catalytic triad
RNA-dependent RNA Polymerase (RdRp)	7BV2	Conessine	−5.737	Ser759, Asp760, Asp761, Asp623, Lys545, Ala762, Ile548, Phe812	H-bonds (Asp623, Lys545), van der Waals (Ala762, Ile548, Phe812), π–alkyl (aromatic residues)
		Nirmatrelvir	−4.676	Asp618, Ser759, Asp760, Asp761, Lys545	Multiple H-bonds (Asp618, Asp760, Asp761), polar (Ser759), electrostatic (Lys545)
Spike RBD	6M0J	Conessine	−7.025	Lys417, Tyr453, Gln493, Leu455, Phe486, Tyr489	H-bonds (Lys417, Tyr453, Gln493), hydrophobic (Leu455, Phe486), π–alkyl (Tyr489)
		Nirmatrelvir	−6.459	Lys417, Tyr449, Glu484, Phe490, Asn501, Tyr505	H-bond (Asn501), polar (Lys417, Glu484), π–π stacking (Tyr505), van der Waals (Tyr449, Phe490)

**Table 4 viruses-17-01435-t004:** Amino Acid Interactions of Conessine with HIV Targets.

HIV Target Protein	Key Amino Acids Interacting with Conessine	Type of Interaction	Functional Significance
Reverse Transcriptase (RT, 3V81)	Tyr181, Tyr188, Trp229, Phe227, Lys101, Val106, Leu100, Val179	Van der Waals, Carbon–H bond, Alkyl	Occupies NNRTI pocket; restricts polymerase conformational dynamics
Protease (PR, 1HVR)	Asp25, Gly27, Asp29, Gly49, Ile50, Val32, Ile84, Pro81, Val82	Van der Waals, Carbon–H bond, Alkyl	Anchors near catalytic dyad; potential inhibition of Gag–Pol polyprotein cleavage
Integrase (IN, 3LPT)	Asp64, Asp116, Glu152, Tyr143, Leu74, Ala128, Val165	Van der Waals, Alkyl, π–alkyl	Disrupts DDE motif and metal coordination; may block DNA strand transfer
gp120–gp41 trimer (4NCO)	Asp368, Ile371, Asn425, Ser375, Asn386, Gly473, Trp427, Phe382	Van der Waals, Carbon–H bond, Alkyl	Engages CD4-binding residues; potential to hinder viral entry and membrane fusion

**Table 5 viruses-17-01435-t005:** Amino acid contacts and interaction types for conessine with selected SARS-CoV-2 targets based on AutoDock Vina docking and 2D interaction analysis.

Target Protein	PDB ID	Key Interacting Residues	Bond Types
Main Protease (M^pro^)	6LU7	His41, Cys145, Gly143, Ser144, Met49, Met165, His164	Carbon–hydrogen bond (Gly143, Ser144), Alkyl (Met49, Met165, His164), π–alkyl (His41)
Papain-like Protease (PL^pro^)	6W9C	Cys111, His272, Leu162, Pro248, Tyr264	Van der Waals (Leu162, Pro248, Tyr264), π–alkyl (Tyr264)
RNA-dependent RNA Polymerase (RdRp)	7BV2	Ser759, Asp760, Asp761, Ala762, Ile548, Phe812, Val557	Van der Waals (Ala762, Ile548, Phe812), Alkyl (Val557)
Spike Protein RBD	6M0J	Lys417, Tyr453, Leu455, Phe486, Tyr489, Gln493	Carbon–hydrogen bond (Lys417, Gln493), π–σ (Tyr453), Alkyl/π–alkyl (Leu455, Phe486, Tyr489)

**Table 6 viruses-17-01435-t006:** Conessine vs. COVID-19 and HIV: Comparative In Silico ADMET, Pharmacokinetic and Safety Profiles with Nirmatrelvir and Doluttegravir.

Domain	Property	Conessine	Nirmatrelvir	Dolutegravir
**Physicochemical**	Molecular Weight (Dalton)	356.60 (55.10%)	499.53 (81.66%)	419.38 (69.72%)
	LogP (log-ratio)	4.81 (86.93%)	1.10 (33.81%)	1.35 (36.95%)
	H-Bond Acceptors	2.00 (16.38%)	5.00 (59.11%)	6.00 (69.50%)
	H-Bond Donors (#)	0.00 (11.77%)	3.00 (77.26%)	2.00 (60.86%)
	Lipinski Rule of 5 (criteria met)	4/4 (63.80%)	4/4 (63.80%)	4/4 (63.80%)
	QED	0.62 (63.47%)	0.50 (47.58%)	0.78 (86.43%)
	Stereo Centers (#)	8.00 (91.97%)	6.00 (87.13%)	2.00 (68.86%)
	TPSA (Å^2^)	6.48 (4.94%)	131.40 (80.85%)	100.87 (67.82%)
**Absorption**	Human Intestinal Absorption	1.00 (53.59%)	0.98 (37.34%)	1.00 (56.15%)
	Oral Bioavailability	0.84 (57.81%)	0.52 (20.86%)	0.91 (76.97%)
	Aqueous Solubility (log mol/L)	−3.68 (37.88%)	−4.34 (25.01%)	−3.94 (33.11%)
	Lipophilicity (log-ratio)	1.62 (52.62%)	2.58 (71.93%)	1.05 (43.04%)
	Hydration Free Energy (kcal/mol)	−3.58 (94.11%)	−8.44 (62.08%)	–12.97 (20.67%)
	Cell Permeability (log10^−6^ cm/s)	−4.88 (56.30%)	−5.26 (33.81%)	−4.78 (62.70%)
	PAMPA Permeability	0.98 (91.55%)	0.83 (57.77%)	0.88 (63.59%)
	P-gp Inhibition	0.33 (67.89%)	0.08 (47.85%)	0.24 (63.82%)
**Distribution**	BBB Penetration	0.96 (79.99%)	0.86 (62.97%)	0.70 (46.41%)
	Plasma Protein Binding (%)	63.70 (35.56%)	81.71 (58.78%)	77.02 (52.31%)
	Vd (Steady State, L/kg)	0.00 (14.85%)	0.71 (42.65%)	0.00 (31.29%)
**Metabolism**	CYP1A2 Inhibition	4.59 × 10^−3^ (24.74%)	3.35 × 10^−4^ (6.09%)	0.01 (37.57%)
	CYP2C19 Inhibition	0.01 (13.61%)	0.06 (40.98%)	0.07 (43.54%)
	CYP2C9 Inhibition	8.90 × 10^−4^ (8.30%)	0.01 (32.92%)	0.11 (66.54%)
	CYP2D6 Inhibition	0.21 (74.53%)	0.02 (36.99%)	0.02 (37.61%)
	CYP3A4 Inhibition	0.02 (38.54%)	0.69 (86.70%)	0.14 (60.37%)
	CYP2C9 Substrate	0.11 (40.52%)	0.03 (11.71%)	0.16 (52.42%)
	CYP2D6 Substrate	0.85 (98.84%)	0.09 (47.15%)	0.04 (26.83%)
	CYP3A4 Substrate	0.56 (56.73%)	0.86 (95.70%)	0.65 (66.54%)
**Excretion**	Half-Life (hr)	0.00 (25.51%)	0.00 (8.76%)	2.97 (42.38%)
	Clearance (Hepatocyte, µL/min/10^6^ cells)	44.11 (57.27%)	13.12 (25.59%)	37.03 (50.79%)
	Clearance (Microsome, µL/min/mg)	0.00 (12.68%)	36.01 (67.24%)	28.34 (60.37%)
**Toxicity/Safety**	hERG Blocking	0.87 (83.95%)	0.56 (67.20%)	0.38 (57.39%)
	Clinical Toxicity	0.16 (61.30%)	0.30 (76.15%)	0.29 (75.53%)
	Mutagenicity	0.02 (8.61%)	0.34 (70.18%)	0.61 (88.60%)
	DILI	0.04 (10.90%)	0.18 (30.79%)	0.94 (86.00%)
	Carcinogenicity	3.40 × 10^−3^ (1.12%)	0.11 (41.53%)	0.10 (37.30%)
	Acute Toxicity LD_50_ (log1/(mol/kg))	2.90 (73.90%)	5.00 (99.96%)	3.16 (85.54%)
	Skin Reaction	0.75 (79.76%)	0.24 (26.79%)	0.15 (13.80%)
	Androgen Receptor (FL)	0.05 (71.19%)	0.02 (38.46%)	0.05 (73.90%)
	Androgen Receptor (LBD)	1.95 × 10^−3^ (16.40%)	0.01 (32.34%)	0.03 (75.34%)
	Aryl Hydrocarbon Receptor	0.01 (25.51%)	1.89 × 10^−3^ (14.66%)	0.02 (50.06%)
	Aromatase	0.01 (41.14%)	0.03 (54.94%)	0.09 (71.00%)
	Estrogen Receptor (FL)	0.08 (46.92%)	0.02 (8.96%)	0.08 (42.77%)
	Estrogen Receptor (LBD)	0.01 (44.51%)	0.01 (42.57%)	0.01 (41.92%)
	PPAR-γ	6.29 × 10^−5^ (7.56%)	1.03 × 10^−3^ (26.25%)	0.01 (58.63%)
	Nrf2/ARE	0.07 (35.28%)	0.19 (59.64%)	0.14 (52.62%)
	ATAD5	9.66 × 10^−5^ (7.06%)	4.50 × 10^−3^ (44.13%)	0.02 (69.87%)
	HSF Response Element	4.03 × 10^−3^ (26.56%)	0.01 (39.28%)	0.01 (39.32%)
	Mitochondrial Potential	0.01 (30.21%)	0.01 (37.18%)	0.03 (48.08%)
	Tumor Protein p53	3.44 × 10^−4^ (6.59%)	0.02 (49.71%)	0.09 (73.94%)

**Table 7 viruses-17-01435-t007:** Predicted protein targets of conessine from SwissTargetPrediction.

Target	Common Name	Uniprot ID	ChEMBL ID	Target Class	Probability *	Known Actives (3D/2D)
Alpha-2a adrenergic receptor	ADRA2A	P08913	CHEMBL1867	Family A G protein-coupled receptor	1	4/1
Adrenergic receptor alpha-2	ADRA2C	P18825	CHEMBL1916	Family A G protein-coupled receptor	1	3/6
Histamine H3 receptor	HRH3	Q9Y5N1	CHEMBL264	Family A G protein-coupled receptor	1	12/21
Serotonin 2a (5-HT2a) receptor	HTR2A	P28223	CHEMBL224	Family A G protein-coupled receptor	0.109339753	52/0
Anti-estrogen binding site (AEBS)	EBP	Q15125	CHEMBL4931	Enzyme	0.109339753	0/2
Lanosterol synthase	LSS	P48449	CHEMBL3593	Enzyme	0.109339753	0/3
Serotonin 6 (5-HT6) receptor	HTR6	P50406	CHEMBL3371	Family A G protein-coupled receptor	0.109339753	136/0
Nociceptin receptor	OPRL1	P41146	CHEMBL2014	Family A G protein-coupled receptor	0.109339753	22/0
Serotonin 7 (5-HT7) receptor	HTR7	P34969	CHEMBL3155	Family A G protein-coupled receptor	0.109339753	17/0
Sonic hedgehog protein (by homology)	SHH	Q15465	CHEMBL5602	Unclassified protein	0.109339753	0/1
Cytochrome P450 19A1	CYP19A1	P11511	CHEMBL1978	Cytochrome P450	0.109339753	0/10
Muscarinic acetylcholine receptor M1	CHRM1	P11229	CHEMBL216	Family A G protein-coupled receptor	0	4/9
Squalene monooxygenase (by homology)	SQLE	Q14534	CHEMBL3592	Enzyme	0	0/2
Sigma opioid receptor	SIGMAR1	Q99720	CHEMBL287	Membrane receptor	0	10/26
Lethal(3)malignant brain tumor-like protein 3	L3MBTL3	Q96JM7	CHEMBL1287623	Reader	0	5/0
Neuronal acetylcholine receptor; alpha4/beta2	CHRNA4 CHRNB2	P43681 P17787	CHEMBL1907589	Ligand-gated ion channel	0	0/7
Serotonin transporter (by homology)	SLC6A4	P31645	CHEMBL228	Electrochemical transporter	0	99/0
Serotonin 2c (5-HT2c) receptor	HTR2C	P28335	CHEMBL225	Family A G protein-coupled receptor	0	33/0
Butyrylcholinesterase	BCHE	P06276	CHEMBL1914	Hydrolase	0	3/9
Serotonin 1b (5-HT1b) receptor	HTR1B	P28222	CHEMBL1898	Family A G protein-coupled receptor	0	13/0
Serotonin 1d (5-HT1d) receptor	HTR1D	P28221	CHEMBL1983	Family A G protein-coupled receptor	0	16/0
Monoamine oxidase A	MAOA	P21397	CHEMBL1951	Oxidoreductase	0	4/0
Monoamine oxidase B	MAOB	P27338	CHEMBL2039	Oxidoreductase	0	3/0
Serotonin 3a (5-HT3a) receptor	HTR3A	P46098	CHEMBL1899	Ligand-gated ion channel	0	8/0
Serotonin 1a (5-HT1a) receptor	HTR1A	P08908	CHEMBL214	Family A G protein-coupled receptor	0	15/0
Nuclear receptor subfamily 1 group D member 2	NR1D2	Q14995	CHEMBL1961784	Nuclear receptor	0	1/0
Dopamine D3 receptor	DRD3	P35462	CHEMBL234	Family A G protein-coupled receptor	0	11/4
Kinesin-like protein 1	KIF11	P52732	CHEMBL4581	Other cytosolic protein	0	23/0
Lethal(3)malignant brain tumor-like protein 1	L3MBTL1	Q9Y468	CHEMBL1287622	Reader	0	1/0
Choline acetylase	CHAT	P28329	CHEMBL4039	Enzyme	0	1/0
Lysine-specific histone demethylase 1	KDM1A	O60341	CHEMBL6136	Eraser	0	3/0
Protein farnesyltransferase	FNTA FNTB	P49354 P49356	CHEMBL2094108	Enzyme	0	3/0
Epoxide hydrolase 1	EPHX1	P07099	CHEMBL1968	Protease	0	0/1
Steroid 5-alpha-reductase 1	SRD5A1	P18405	CHEMBL1787	Oxidoreductase	0	0/62
Thromboxane-A synthase	TBXAS1	P24557	CHEMBL1835	Cytochrome P450	0	0/1
Steroid 5-alpha-reductase 2	SRD5A2	P31213	CHEMBL1856	Oxidoreductase	0	0/107
Cytochrome P450 17A1	CYP17A1	P05093	CHEMBL3522	Cytochrome P450	0	0/11
Histamine H4 receptor	HRH4	Q9H3N8	CHEMBL3759	Family A G protein-coupled receptor	0	7/0
Dipeptidyl peptidase IV	DPP4	P27487	CHEMBL284	Protease	0	4/0
Phospholipase A2 group 1B (by homology)	PLA2G1B	P04054	CHEMBL4426	Enzyme	0	3/0
Delta opioid receptor	OPRD1	P41143	CHEMBL236	Family A G protein-coupled receptor	0	9/3
Serine/threonine-protein kinase AKT2	AKT2	P31751	CHEMBL2431	Kinase	0	15/0
Serine/threonine-protein kinase AKT	AKT1	P31749	CHEMBL4282	Kinase	0	17/0
Nitric-oxide synthase, brain	NOS1	P29475	CHEMBL3568	Enzyme	0	12/0
Opioid growth factor receptor-like protein 1	OGFRL1	Q5TC84	CHEMBL3638334	Unclassified protein	0	1/0
Mu opioid receptor	OPRM1	P35372	CHEMBL233	Family A G protein-coupled receptor	0	16/4
Alpha-1a adrenergic receptor (by homology)	ADRA1A	P35348	CHEMBL229	Family A G protein-coupled receptor	0	7/0
Myeloperoxidase	MPO	P05164	CHEMBL2439	Enzyme	0	5/0
Acetylcholinesterase	ACHE	P22303	CHEMBL220	Hydrolase	0	6/7
Geranylgeranyl transferase type I	PGGT1B FNTA	P53609 P49354	CHEMBL2095164	Enzyme	0	1/0
Nitric oxide synthase, inducible (by homology)	NOS2	P35228	CHEMBL4481	Enzyme	0	9/0
Nitric-oxide synthase, endothelial	NOS3	P29474	CHEMBL4803	Enzyme	0	2/0
Serotonin 2b (5-HT2b) receptor	HTR2B	P41595	CHEMBL1833	Family A G protein-coupled receptor	0	5/0
Alpha-1d adrenergic receptor	ADRA1D	P25100	CHEMBL223	Family A G protein-coupled receptor	0	2/0
Niemann-Pick C1-like protein 1	NPC1L1	Q9UHC9	CHEMBL2027	Other membrane protein	0	0/2
Serine/threonine-protein kinase PIM1	PIM1	P11309	CHEMBL2147	Kinase	0	3/0
LSD1/CoREST complex	RCOR1 KDM1A	Q9UKL0 O60341	CHEMBL3137262	Eraser	0	1/0
Neuropilin-1	NRP1	O14786	CHEMBL5174	Secreted protein	0	1/0
Epoxide hydratase	EPHX2	P34913	CHEMBL2409	Protease	0	0/2
Neurokinin 1 receptor (by homology)	TACR1	P25103	CHEMBL249	Family A G protein-coupled receptor	0	1/0
Gamma-secretase	PSEN2 PSENEN NCSTN APH1A PSEN1 APH1B	P49810 Q9NZ42 Q92542 Q96BI3 P49768 Q8WW43	CHEMBL2094135	Protease	0	2/0
Anandamide amidohydrolase	FAAH	O00519	CHEMBL2243	Enzyme	0	0/8
Monoglyceride lipase	MGLL	Q99685	CHEMBL4191	Enzyme	0	0/3
Telomerase reverse transcriptase	TERT	O14746	CHEMBL2916	Enzyme	0	1/0
Amiloride-sensitive cation channel 3	ASIC3	Q9UHC3	CHEMBL5368	Ligand-gated ion channel	0	2/0
Beta-glucocerebrosidase	GBA	P04062	CHEMBL2179	Enzyme	0	0/1
Beta-glucosidase	GBA2	Q9HCG7	CHEMBL3761	Enzyme	0	0/2
Voltage-gated T-type calcium channel alpha-1G subunit	CACNA1G	O43497	CHEMBL4641	Voltage-gated ion channel	0	1/0
Dopamine D2 receptor	DRD2	P14416	CHEMBL217	Family A G protein-coupled receptor	0	27/8
Inhibitor of apoptosis protein 3	XIAP	P98170	CHEMBL4198	Other cytosolic protein	0	1/0
Peroxisome proliferator-activated receptor alpha	PPARA	Q07869	CHEMBL239	Nuclear receptor	0	0/1
Vanilloid receptor	TRPV1	Q8NER1	CHEMBL4794	Voltage-gated ion channel	0	0/3
Glucose-dependent insulinotropic receptor	GPR119	Q8TDV5	CHEMBL5652	Family A G protein-coupled receptor	0	0/1
Dopamine D4 receptor	DRD4	P21917	CHEMBL219	Family A G protein-coupled receptor	0	16/2
Cyclin-dependent kinase 9	CDK9	P50750	CHEMBL3116	Kinase	0	1/0
Urotensin II receptor	UTS2R	Q9UKP6	CHEMBL3764	Family A G protein-coupled receptor	0	1/0
Dopamine D5 receptor	DRD5	P21918	CHEMBL1850	Family A G protein-coupled receptor	0	1/0
Dopamine D1 receptor	DRD1	P21728	CHEMBL2056	Family A G protein-coupled receptor	0	2/0
11-beta-hydroxysteroid dehydrogenase 2	HSD11B2	P80365	CHEMBL3746	Enzyme	0	0/9
Serotonin 1f (5-HT1f) receptor	HTR1F	P30939	CHEMBL1805	Family A G protein-coupled receptor	0	2/0
Adenosine A3 receptor	ADORA3	P0DMS8	CHEMBL256	Family A G protein-coupled receptor	0	1/0
Kinesin-1 heavy chain/Tyrosine-protein kinase receptor RET	RET	P07949	CHEMBL2041	Kinase	0	1/0
Kappa Opioid receptor	OPRK1	P41145	CHEMBL237	Family A G protein-coupled receptor	0	6/1
Anti-estrogen binding site (AEBS)	DHCR7 EBP	Q9UBM7 Q15125	CHEMBL612409	Enzyme	0	0/1
Serine/threonine-protein kinase PIM2	PIM2	Q9P1W9	CHEMBL4523	Kinase	0	1/0
Squalene synthetase (by homology)	FDFT1	P37268	CHEMBL3338	Enzyme	0	0/3
LXR-alpha	NR1H3	Q13133	CHEMBL2808	Nuclear receptor	0	0/2
Polyadenylate-binding protein 1	PABPC1	P11940	CHEMBL1293286	Unclassified protein	0	0/1
Prolyl endopeptidase	PREP	P48147	CHEMBL3202	Protease	0	0/4
Cholesteryl ester transfer protein	CETP	P11597	CHEMBL3572	Other ion channel	0	0/1
Sphingosine kinase 1	SPHK1	Q9NYA1	CHEMBL4394	Enzyme	0	0/1
Cytochrome P450 2C9	CYP2C9	P11712	CHEMBL3397	Cytochrome P450	0	0/1
Androgen Receptor	AR	P10275	CHEMBL1871	Nuclear receptor	0	0/4
Metastin receptor	KISS1R	Q969F8	CHEMBL5413	Family A G protein-coupled receptor	0	0/1
Protein-tyrosine phosphatase 1B	PTPN1	P18031	CHEMBL335	Phosphatase	0	0/3
Cytochrome P450 24A1	CYP24A1	Q07973	CHEMBL4521	Enzyme	0	0/1
Mineralocorticoid receptor	NR3C2	P08235	CHEMBL1994	Nuclear receptor	0	0/1
Cannabinoid receptor 1	CNR1	P21554	CHEMBL218	Family A G protein-coupled receptor	0	3/79
Tyrosine-protein kinase FYN	FYN	P06241	CHEMBL1841	Kinase	0	0/1
Epidermal growth factor receptor erbB1	EGFR	P00533	CHEMBL203	Kinase	0	0/1

## Data Availability

The original contributions presented in this study are included in the article/Appendix A. Further inquiries can be directed to the corresponding author(s).

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
