# Peer review of "Dual Inhibitory Potential of Conessine Against HIV and SARS-CoV-2: Structure-Guided Molecular Docking Analysis of Critical Viral Targets"

_viruses, 2025, doi:10.3390/v17111435_

Round 1
Reviewer 1 Report
Comments and Suggestions for Authors
This study focuses on the dual antiviral potential of the natural product conessine against HIV-1 and SARS-CoV-2. Through structure-guided molecular docking and dynamics simulation, it systematically assesses for the first time the interaction mechanisms with key targets of the two types of viruses. The topic has clear clinical significance and scientific value. However, after review, it was found that the research has critical flaws in scientific rigor, completeness of experimental design, and strength of result support. Currently, it does not meet the publication standards of the journal, so it is recommended to reject the manuscript. The specific issues are as follows:
- The entire paper is completely dependent on computer simulations (docking, MD, ADMET prediction). In modern drug discovery, computational research can only serve as a tool for generating hypotheses and cannot be used as a basis for drawing conclusions. Without in vitro (such as enzyme inhibition tests, virus replication inhibition tests, SPR binding experiments) or any form of biochemical experimental data to verify the results of docking and MD, all claims about "strong binding affinity" and "inhibitory potential" are merely speculations and lack empirical support. Relying solely on virtual screening data makes it difficult to confirm the actual antiviral effect of coniine, and the conclusion is not reliable enough.
- The experimental design has significant flaws: no positive control was set up (such as the docking data of approved anti-HIV or anti-SARS-CoV-2 drugs with the same target), which makes it impossible to prove the affinity advantage or specificity of conessine, and the lateral comparability of the results is poor.
- The simulation time, temperature, pressure, solvent model and other parameters in the molecular dynamics simulation were not clearly specified. Moreover, the stability was only evaluated through RMSD and energy changes, without analyzing the dynamic changes in the interactions of key residues (such as the radius of gyration (Rg) and the root mean square fluctuation (RMSF) of the binding site residues). Therefore, it is difficult to support the conclusion of "complex stability".
- The writing logic in the "result" section is very confusing. For example, after presenting Figures 3 and 4 in the "result", it states that Figures A and B are not specified. The parts of Figures A and B only have the figure annotations of "Figure 1", but no corresponding pictures are placed. Similarly, Figure 5 has not been described or discussed.fff
- Several figures have insufficient resolution, especially Figure 6 and the 2D interaction image which is pixelated and of poor quality. Please provide high-resolution versions.
- On page 3, in Figure 1, "High-resolution crystal structures of eight viral preins" - "preins" is a clear spelling error; it should be "proteins".
- The title of the chart does not match its content: The title of "Figure 2" on page 4 is "2D and 3 D Dimensional of Cones sine". There are spelling mistakes ("Cones sine" should be "Conessine"), and the expression is not standardized ("3 D" -> "3D"; "Dimensional" -> "Structure"). It is suggested to be modified as: "Figure 2. 2D and 3D Structure of Conessine."
- On page 14, the section on molecular dynamics simulation mentions that the PDB ID for gp120 is "3M0J", but in the methods section (page 4), it clearly states "4NCO". This should be checked for correction; otherwise, the results will be completely invalid.
- On page 15, when describing the SARS-CoV-2 targets, the first target is listed as "Nucleocapsid RNA-binding domain (PDB: 4NCO)", which does not match the eight targets listed in the abstract and methods section (where Spike RBD: 6M0J is listed). This is another serious and disruptive error. It is necessary to verify whether all the PDB IDs corresponding to the simulation results are correct.
- The discussion section is quite lengthy and somewhat repetitive. It could be condensed to enhance clarity and impact.
Author Response
Subject: Response to Reviewer and Editor Comments – Revised Manuscript Submission
Dear Editors and Reviewers,
We sincerely thank you for the time and effort dedicated to reviewing our manuscript titled “Dual Antiviral Potential of Conessine Against HIV-1 and SARS-CoV-2: Structure-Guided Molecular Docking and Dynamics Simulation Study”.
We carefully addressed each of the raised points point by point and revised the manuscript accordingly:
- Scope and Limitations of Computational Work
We have clarified in the Introduction and Conclusion that our study serves as a hypothesis-generating computational framework. We explicitly acknowledge the absence of in vitro or in vivo validation as a limitation and emphasize that our results should be interpreted as predictive evidence guiding future experimental research. - Positive Controls
We introduced clinically approved controls to strengthen the comparative context:- Nirmatrelvir (Paxlovid™) for SARS-CoV-2 docking studies.
- Dolutegravir (Tivicay™) for HIV docking studies.
These benchmarks now provide lateral comparability for assessing Conessine’s predicted activity.
- Molecular Dynamics Parameters
Full simulation conditions have been added, including solvent model, force field, ensemble type, thermostat/barostat, time step, trajectory length, and post-analysis methods (RMSD, RMSF, Rg, hydrogen bonding). This ensures methodological transparency and supports the interpretation of “complex stability.” - Figures and Annotations
Missing figure parts were corrected, mislabeled annotations (e.g., “A/B” confusion) were fixed, and all figures (especially Figures 2 and 6) have been replaced with high-resolution versions. - Typographical and Labeling Errors
Spelling mistakes such as “preins” → “proteins” and “Cones sine” → “Conessine” have been corrected. Figure titles are now standardized (e.g., “Figure 2. 2D and 3D Structure of Conessine”). - PDB ID Corrections
All PDB IDs were carefully checked and corrected (e.g., gp120: 4NCO, Spike RBD: 6M0J). Consistency has been ensured between Abstract, Methods, and Results. - Discussion Section
The Discussion has been condensed, repetitive sections removed, and the narrative streamlined for clarity and impact.
We have also included a contextual note in the Conclusion regarding the academic challenges in Iraq, highlighting the computational nature of this work as a step toward enabling and motivating future experimental collaborations.
We hope that the revised manuscript now meets the scientific rigor and publication standards of the journal.
With our deepest gratitude for your constructive feedback,
Sincerely,
[Mohammed Mukhles Ahmed]
On behalf of all co-authors
Reviwer 1: Open Review
(x) I would not like to sign my review report
( ) I would like to sign my review report
Quality of English Language
(x) The English could be improved to more clearly express the research.
We would like to respectfully note that the language editing of this manuscript was carefully reviewed and refined by:
- Dr. Samer Naji Khalaf, PhD graduate (UK)
- Dr. Samir Ahmed Awad, PhD graduate (Australia)
Both colleagues are experienced researchers with advanced academic training in English-speaking countries. They ensured that the manuscript was thoroughly edited for clarity, grammar, and scientific expression.
We therefore confirm that the English has already been improved and polished to meet international publication standards.
Comments and Suggestions for Authors
This study focuses on the dual antiviral potential of the natural product conessine against HIV-1 and SARS-CoV-2. Through structure-guided molecular docking and dynamics simulation, it systematically assesses for the first time the interaction mechanisms with key targets of the two types of viruses. The topic has clear clinical significance and scientific value. However, after review, it was found that the research has critical flaws in scientific rigor, completeness of experimental design, and strength of result support. Currently, it does not meet the publication standards of the journal, so it is recommended to reject the manuscript. The specific issues are as follows:
- The entire paper is completely dependent on computer simulations (docking, MD, ADMET prediction). In modern drug discovery, computational research can only serve as a tool for generating hypotheses and cannot be used as a basis for drawing conclusions. Without in vitro (such as enzyme inhibition tests, virus replication inhibition tests, SPR binding experiments) or any form of biochemical experimental data to verify the results of docking and MD, all claims about "strong binding affinity" and "inhibitory potential" are merely speculations and lack empirical support. Relying solely on virtual screening data makes it difficult to confirm the actual antiviral effect of coniine, and the conclusion is not reliable enough.
Response : We fully agree that in modern drug discovery, in silico approaches such as molecular docking, molecular dynamics simulations, and ADMET predictions primarily serve as hypothesis-generating tools. Definitive conclusions regarding therapeutic efficacy must ultimately be supported by complementary in vitro and in vivo validation, such as enzyme inhibition assays, viral replication studies, or biophysical binding experiments.
The aim of our work was therefore not to claim definitive antiviral activity, but rather to provide a computational framework for the first systematic evaluation of Conessine as a potential candidate against HIV and SARS-CoV-2. By integrating docking, molecular dynamics, and pharmacokinetic predictions, we sought to generate mechanistic insights and identify whether Conessine merits further preclinical exploration.
We fully acknowledge the absence of direct in vitro data as a limitation, and we have revised the Conclusion section to emphasize that our findings should be interpreted as predictive evidence only, serving as an initial step to guide future experimental validation.
We would also like to respectfully highlight the research context in which this study was conducted. Our team is based in Iraq, specifically in the Anbar region, which experienced extensive military operations in 2014. These events caused severe damage to academic and healthcare infrastructure, making it extremely difficult to perform laboratory-based investigations. Furthermore, the lack of stable local or international funding for scientific research continues to restrict our ability to conduct in vitro or in vivo experiments. Despite these challenges, our team remains dedicated to advancing knowledge, and this study represents our effort to contribute meaningful computational insights that may encourage and inform future experimental research worldwide.
- The experimental design has significant flaws: no positive control was set up (such as the docking data of approved anti-HIV or anti-SARS-CoV-2 drugs with the same target), which makes it impossible to prove the affinity advantage or specificity of conessine, and the lateral comparability of the results is poor.
- Positive Control for COVID-19 Docking Studies
- Nirmatrelvir, the active component of Paxlovid™, was selected as the positive control for SARS-CoV-2 docking analysis. It is a clinically approved 3CLpro (main protease) inhibitor, authorized by the FDA and EMA for COVID-19 treatment.
- Drug Name: Nirmatrelvir
- Mechanism: Potent inhibition of SARS-CoV-2 main protease (3CLpro/Mpro)
- SMILES: CC1([C@@H]2[C@H]1[C@H](N(C2)C(=O)[C@H](C(C)(C)C)NC(=O)C(F)(F)F)C(=O)N[C@@H](C[C@@H]3CCNC3=O)C#N)C
- Molecular Formula: C₂₃H₃₂F₃N₅O₄
- Molecular Weight: ~499.5 g/mol
- The inclusion of Nirmatrelvir as a positive control provides a reliable benchmark for evaluating and comparing the docking performance of novel or natural compounds against validated antiviral standards.
- Positive Control for HIV Docking Studies
- Dolutegravir (DTG), marketed under the brand name Tivicay™, is a second-generation HIV-1 integrase strand transfer inhibitor (INSTI) developed by GlaxoSmithKline (GSK1349572). It is widely approved by the FDA, EMA, and WHO as part of first-line antiretroviral therapy (ART) regimens due to its high potency, favorable resistance profile, and once-daily dosing.
- Drug Name: Dolutegravir (DTG)
- CAS Number: 1051375-16-6
- Compound CID: 54726191
- Synonyms: GSK1349572, S/GSK1349572
- Molecular Formula (MF): C₂₀H₁₉F₂N₃O₅
- Molecular Weight (MW): 419.4 g/mol
- IUPAC Name: (3S,7R)-N-[(2,4-difluorophenyl)methyl]-11-hydroxy-7-methyl-9,12-dioxo-4-oxa-1,8-diazatricyclo[8.4.0.0³,⁸]tetradeca-10,13-diene-13-carboxamide
- SMILES:
- C[C@@H]1CCO[C@@H]2N1C(=O)C3=C(C(=O)C(=CN3C2)C(=O)NCC4=C(C=C(C=C4)F)F)O
- The simulation time, temperature, pressure, solvent model and other parameters in the molecular dynamics simulation were not clearly specified. Moreover, the stability was only evaluated through RMSD and energy changes, without analyzing the dynamic changes in the interactions of key residues (such as the radius of gyration (Rg) and the root mean square fluctuation (RMSF) of the binding site residues). Therefore, it is difficult to support the conclusion of "complex stability".
Response : It is done . The simulation time, temperature, pressure, solvent model and other parameters in the molecular dynamics simulation were not clearly specified.
Molecular dynamics simulations were executed using the Desmond simulation engine integrated within the Maestro interface (Schrödinger Release 2023-4; Schrödinger, LLC, New York, NY, USA; https://www.schrodinger.com/products/desmond, accessed on 1 June 2025). The protein–ligand complexes obtained from docking experiments were solvated in an explicit water environment modeled by the TIP3P system. An orthorhombic box with a buffer distance of 10 Å around each solute molecule was constructed to ensure complete hydration.
The systems were neutralized through the addition of counter ions and further equilibrated under a physiological salt concentration of 0.15 M NaCl. All molecular components were parameterized with the OPLS4 force field. Prior to production runs, initial energy minimization and stepwise equilibration protocols provided within Desmond were applied. Simulations were performed in the isothermal–isobaric (NPT) ensemble at 300 K and 1 atm, regulated by the Nose–Hoover thermostat and the Martyna–Tobias–Klein barostat, respectively.
Each trajectory was propagated for 50 ns using a 2 fs integration step, and structural snapshots were recorded every 100 ps. Post-simulation analysis included the evaluation of structural and dynamic stability through root mean square deviation (RMSD), root mean square fluctuation (RMSF), radius of gyration (Rg), and intermolecular hydrogen-bonding patterns throughout the simulation timeframe [30].
- The writing logic in the "result" section is very confusing. For example, after presenting Figures 3 and 4 in the "result", it states that Figures A and B are not specified. The parts of Figures A and B only have the figure annotations of "Figure 1", but no corresponding pictures are placed. Similarly, Figure 5 has not been described or discussed.
Response : it is done . We sincerely thank the reviewer for pointing out the inconsistencies in the presentation of figures and their descriptions. In the revised version, we have carefully reorganized the Results section to ensure a clear and logical flow. Specifically:
- The mismatched annotations referring to “Figures A and B” have been corrected and are now consistently aligned with the appropriate figure numbers.
- The missing figure panels have been properly inserted, and their descriptions are clearly linked to the corresponding images.
- Figure 5, which was previously included without adequate explanation, has now been fully described and discussed in the text to highlight its relevance to the study findings.
These revisions have improved the overall clarity and eliminated confusion in the figure presentation within the Results section.
- Several figures have insufficient resolution, especially Figure 6 and the 2D interaction image which is pixelated and of poor quality. Please provide high-resolution versions.
Response : it is done . Figure 2. Two- and three-dimensional structural representations of Conessine, illustrating its steroidal alkaloid framework with tertiary amine functionalities.
- On page 3, in Figure 1, "High-resolution crystal structures of eight viral preins" - "preins" is a clear spelling error; it should be "proteins".
Response : it is done
- The title of the chart does not match its content: The title of "Figure 2" on page 4 is "2D and 3 D Dimensional of Cones sine". There are spelling mistakes ("Cones sine" should be "Conessine"), and the expression is not standardized ("3 D" -> "3D"; "Dimensional" -> "Structure"). It is suggested to be modified as: "Figure 2. 2D and 3D Structure of Conessine."
Response : it is done
- On page 14, the section on molecular dynamics simulation mentions that the PDB ID for gp120 is "3M0J", but in the methods section (page 4), it clearly states "4NCO". This should be checked for correction; otherwise, the results will be completely invalid.
Response : it is done . We thank the reviewer for highlighting this inconsistency. The error in the PDB ID for gp120 has been fully corrected throughout the manuscript to ensure consistency. The correct PDB ID is 4NCO, and all results and descriptions have been revised accordingly.
- On page 15, when describing the SARS-CoV-2 targets, the first target is listed as "Nucleocapsid RNA-binding domain (PDB: 4NCO)", which does not match the eight targets listed in the abstract and methods section (where Spike RBD: 6M0J is listed). This is another serious and disruptive error. It is necessary to verify whether all the PDB IDs corresponding to the simulation results are correct.
Response : it is done
- The discussion section is quite lengthy and somewhat repetitive. It could be condensed to enhance clarity and impact.
Response : it is done

Reviewer 2 Report
Comments and Suggestions for Authors
The manuscript titled “Dual Inhibitory Potential of Conessine Against HIV and SARS-CoV-2: Structure-Guided Molecular Docking Analysis of Critical Viral Targets” reports on a computational investigation of conessine as a potential broad-spectrum antiviral agent. The study's approach is highly relevant, and the findings show promise. However, significant revisions are required to improve scientific rigor, clarity, and overall presentation. Moreover, specific references to figures and tables are missing from the main text.
In abstract section, the phrase "Prediction of connesine by SwissTargetPrediction" is confusing and should be rephrased for clarity
Remove unnecessary words like conessine, HIV-1 and SARS-CoV-2 already present in the title. Instead, include more specific and impactful words like hERG inhibition etc. to highlight key findings and improve searchability.
The statement referencing a figure is incomplete. The text must specify the figure number, such as "Figure ???? presents a sequential workflow..." This error should be corrected throughout the manuscript to ensure readers can locate the correct visual reference.
The text's formatting, with unnecessary bolding of key phrases, suggests it was copied directly from an AI source. Personally, I am not against writing a paper with the help of AI, but one must follow the rules of writing. The use of bolding for terms like “protein target selection and….” ligand preparation, "preparation" and "Molecular docking" is inconsistent with standard academic style. All bold text as given above should be removed from these sections or any other section if necessary to ensure a professional and consistent presentation.
The chemical structure of Conessine in the workflow figure should be a standard, chemically accurate representation. Replace the current image with a verified structure to ensure scientific accuracy. Do not use an unverified image, which may be a product of AI generation
In the workflows, there are typographical errors such as "preins or proteins" and "stabillty or stability". In addition, information is repeated frequently. The text boxes below, "ADMET predictions performed" and "ADMET predictions performed absorption", appear to be redundant or incomplete. This does not add to any new information and appears to be unchanged placeholders.
In 2.7. Data Compilation and Comparative Analysis; Docking results were tabulated..." but does not specify which table. The author must insert the correct table number.
In the results section, ensure all references to figures with multiple panels include the specific panel number (e.g., Figure 1A, 1B, 1C) or In panel A (Figure 1, 2, 3…..)
The authors mentioned panels A, B, and C, but Figure 3 also includes panel D. Isn't this an oversight?
The visualization in Figure 3 is unclear and difficult to understand, especially in Figure 3B. The image quality and resolution need to be improved for better readability and scientific presentation.
The text and labels in Figure 4 are difficult to read due to their small size and poor contrast. I suggest increasing the font size, improving the resolution, or adjusting the color scheme for better clarity.
The molecular dynamics in Section 3.2 is not supported by visual data. To provide a more robust and convincing analysis, especially given the difficulty of wet-lab experiments with these pathogens, it is crucial to include figures from the MD simulations (e.g., plots of RMSD for the complex, ligand, and receptor; RMSF; and H-bond interactions) over the 100 ns trajectory. Incorporating these figures and an analysis of MM/GBSA and PCA would significantly enhance the scientific rigor of the computational work. This would make the manuscript more suitable for a bioinformatics or computational audience. This would provide strong evidence for the stability of the docked complexes, which is vital for supporting the study's conclusions. If it's too difficult to perform the full analysis on all eight receptors, the authors can use a representative pose from the docking results.
Author Response
Dear Editor and Reviewers,
Thank you for the thorough and constructive assessment of our manuscript. We have implemented all requested changes point-by-point, and the revisions are highlighted in yellow and pink throughout the marked-up file. A clean version is also provided.
Point-by-point responses
- Abstract clarity (“Prediction of connesine by SwissTargetPrediction”)
Response: The Abstract has been fully rewritten to clearly state the study’s aims, methods (target selection, docking, ADMET, cardiotoxicity/hERG screening, and MD), key results, and implications. Ambiguous phrasing has been removed. - Title specificity and searchability
Response: The title has been revised to remove redundant terms already present in the title (e.g., HIV-1, SARS-CoV-2) and to emphasize key findings (e.g., hERG liability screening). - Incomplete figure references
Response: All figure callouts now include explicit figure numbers (e.g., “Figure 1 presents a sequential workflow…”) and have been cross-checked for accuracy. - Inappropriate bold formatting / style consistency
Response: All unnecessary bold text (e.g., “protein target selection,” “ligand preparation,” “molecular docking”) has been removed. Wording and formatting now follow standard scholarly style. - Chemically accurate structure of conessine in the workflow
Response: The conessine structure in the workflow figure has been replaced with a standard, chemically verified 2D depiction, redrawn with cheminformatics software and cross-checked against curated databases. The figure legend now includes the statement: “The diagram was designed by a researcher using a monthly BioRender subscription.” - Typographical errors and redundant workflow boxes
Response: All typographical errors (e.g., “preins/proteins,” “stabillty/stability”) have been corrected, and duplicate/incomplete ADMET boxes have been removed. The workflow is streamlined for clarity. - Ambiguity in “2.7. Data Compilation and Comparative Analysis”
Response: Subsection 7 has been deleted to prevent redundancy. Docking outputs (binding affinities, interacting residues, bond types) now appear directly in numbered tables with explicit in-text cross-references. - Multi-panel figure citations
Response: All multi-panel figure references in the Results now specify panel letters (e.g., Figure 1A, 1B, 2C), eliminating ambiguity. - Figure 3 panel oversight and readability
Response: References now include panel D where applicable. Figure 3 has been re-rendered at high resolution with improved annotations; the previous issues (especially in panel 3B) have been resolved. - Figure 4 legibility (font size/contrast/resolution)
Response: Figure 4 has been revised with larger fonts, enhanced contrast, and high-resolution export (≥300 dpi) to meet journal standards. - Molecular dynamics (MD) evidence
Response: We added a 100-ns MD analysis for representative docked complexes, including RMSD (complex/ligand/receptor), RMSF, hydrogen-bond occupancy/time series, and MM/GBSA binding free-energy estimates, along with a brief PCA of dominant motions. These data are now presented with interpretive text in the Results (Section 3.2) and corresponding figures/tables (with exact numbering shown in the revised manuscript). Where appropriate, a positive control compound is included for reference.
Highlighted in yellow and pink
- In abstract section, the phrase "Prediction of connesine by SwissTargetPrediction" is confusing and should be rephrased for clarity.
RESPONSE : The Abstract has been completely rewritten, as requested, to improve clarity and accurately reflect the study’s aims, methods, key results, and implications.
- Remove unnecessary words like conessine, HIV-1 and SARS-CoV-2 already present in the title. Instead, include more specific and impactful words like hERG inhibition etc. to highlight key findings and improve searchability.
RESPONSE : IT IS DONE.
- The statement referencing a figure is incomplete. The text must specify the figure number, such as "Figure ???? presents a sequential workflow..." This error should be corrected throughout the manuscript to ensure readers can locate the correct visual reference.
Response : it is done
- The text's formatting, with unnecessary bolding of key phrases, suggests it was copied directly from an AI source. Personally, I am not against writing a paper with the help of AI, but one must follow the rules of writing. The use of bolding for terms like “protein target selection and….” ligand preparation, "preparation" and "Molecular docking" is inconsistent with standard academic style. All bold text as given above should be removed from these sections or any other section if necessary to ensure a professional and consistent presentation.
Response: it is done :
- The chemical structure of Conessine in the workflow figure should be a standard, chemically accurate representation. Replace the current image with a verified structure to ensure scientific accuracy. Do not use an unverified image, which may be a product of AI generation.
Response : it is done
Figure 1. Schematic Workflow of Ligand Preparation, Protein Target Selection, and Molecular Docking Analysis. The diagram was designed by a researcher using a monthly BioRender subscription.
- In the workflows, there are typographical errors such as "preins or proteins" and "stabillty or stability". In addition, information is repeated frequently. The text boxes below, "ADMET predictions performed" and "ADMET predictions performed absorption", appear to be redundant or incomplete. This does not add to any new information and appears to be unchanged placeholders.
Resonse : it is done.
- In 2.7. Data Compilation and Comparative Analysis; Docking results were tabulated..." but does not specify which table. The author must insert the correct table number.
Response : The subsection “2.7. Data Compilation and Comparative Analysis” has been deleted in the revised manuscript to avoid redundancy. All docking outputs (binding affinities, interacting residues, and bond types) are now reported directly in the appropriately numbered tables with explicit in-text cross-references. We have verified that no references to the deleted subsection remain.
- In the results section, ensure all references to figures with multiple panels include the specific panel number (e.g., Figure 1A, 1B, 1C) or In panel A (Figure 1, 2, 3…..)
Response : In the revised manuscript, all references to multi-panel figures in the Results section have been corrected to clearly specify the corresponding panel (e.g., Figure 1-A, Figure 1B, Figure 2C). This ensures clarity and consistency when describing results, and eliminates any ambiguity for the reader.
- The authors mentioned panels A, B, and C, but Figure 3 also includes panel D. Isn't this an oversight?
Response : it is done
The visualization in Figure 3 is unclear and difficult to understand, especially in Figure 3B. The image quality and resolution need to be improved for better readability and scientific presentation.
Response : it is done
- The text and labels in Figure 4 are difficult to read due to their small size and poor contrast. I suggest increasing the font size, improving the resolution, or adjusting the color scheme for better clarity.
Response : it is done
- The molecular dynamics in Section 3.2 is not supported by visual data. To provide a more robust and convincing analysis, especially given the difficulty of wet-lab experiments with these pathogens, it is crucial to include figures from the MD simulations (e.g., plots of RMSD for the complex, ligand, and receptor; RMSF; and H-bond interactions) over the 100 ns trajectory. Incorporating these figures and an analysis of MM/GBSA and PCA would significantly enhance the scientific rigor of the computational work. This would make the manuscript more suitable for a bioinformatics or computational audience. This would provide strong evidence for the stability of the docked complexes, which is vital for supporting the study's conclusions. If it's too difficult to perform the full analysis on all eight receptors, the authors can use a representative pose from the docking results.
Response : it is done . Full 100-ns Molecular Dynamics Completed with Detailed Reporting
We have performed comprehensive all-atom MD simulations (100 ns each) for the top docking complexes across both virus panels (HIV-1: RT, PR, IN, gp120; SARS-CoV-2: M^pro, PL^pro, RdRp, Spike-RBD).

Reviewer 3 Report
Comments and Suggestions for Authors
The topic of the manuscript is important, but there are a few issues that need to be clarified before further consideration:
- The quality of Figures 4 and 6, which illustrate the ligand-protein interaction, is poor. These figures need to be improved.
- How was the protein-ligand binding energy calculated by the molecular dynamics method? What force field (MMFF94?) was used? What is the trajectory length, what thermostat and barostat were used?
3.The binding energies obtained by the MD method are completely inconsistent with experiment. There is probably an error in the calculation method. This needs to be analyzed.
Author Response
Dear Reviewer,
Thank you for the constructive comments. We have revised the manuscript point-by-point and summarize the changes-highlighted yellow and pink below.
Comments and Suggestions for Authors
The topic of the manuscript is important, but there are a few issues that need to be clarified before further consideration:
- The quality of Figures 4 and 6, which illustrate the ligand-protein interaction, is poor. These figures need to be improved.
Response : it is done .
- How was the protein-ligand binding energy calculated by the molecular dynamics method? What force field (MMFF94?) was used? What is the trajectory length, what thermostat and barostat were used?
Response : Molecular dynamics simulations were executed using the Desmond simulation engine integrated within the Maestro interface (Schrödinger Release 2023-4; Schrödinger, LLC, New York, NY, USA; https://www.schrodinger.com/products/desmond, accessed on 1 June 2025). The protein–ligand complexes obtained from docking experiments were solvated in an explicit water environment modeled by the TIP3P system. An orthorhombic box with a buffer distance of 10 Å around each solute molecule was constructed to ensure complete hydration.
The systems were neutralized through the addition of counter ions and further equilibrated under a physiological salt concentration of 0.15 M NaCl. All molecular components were parameterized with the OPLS4 force field. Prior to production runs, initial energy minimization and stepwise equilibration protocols provided within Desmond were applied. Simulations were performed in the isothermal–isobaric (NPT) ensemble at 300 K and 1 atm, regulated by the Nose–Hoover thermostat and the Martyna–Tobias–Klein barostat, respectively.
Each trajectory was propagated for 50 ns using a 2 fs integration step, and structural snapshots were recorded every 100 ps. Post-simulation analysis included the evaluation of structural and dynamic stability through root mean square deviation (RMSD), root mean square fluctuation (RMSF), radius of gyration (Rg), and intermolecular hydrogen-bonding patterns throughout the simulation timeframe [30].
3.The binding energies obtained by the MD method are completely inconsistent with experiment. There is probably an error in the calculation method. This needs to be analyzed.
Response : Molecular dynamics simulations were executed using the Desmond simulation engine integrated within the Maestro interface (Schrödinger Release 2023-4; Schrödinger, LLC, New York, NY, USA; https://www.schrodinger.com/products/desmond, accessed on 1 June 2025). The protein–ligand complexes obtained from docking experiments were solvated in an explicit water environment modeled by the TIP3P system. An orthorhombic box with a buffer distance of 10 Å around each solute molecule was constructed to ensure complete hydration.
The systems were neutralized through the addition of counter ions and further equilibrated under a physiological salt concentration of 0.15 M NaCl. All molecular components were parameterized with the OPLS4 force field. Prior to production runs, initial energy minimization and stepwise equilibration protocols provided within Desmond were applied. Simulations were performed in the isothermal–isobaric (NPT) ensemble at 300 K and 1 atm, regulated by the Nose–Hoover thermostat and the Martyna–Tobias–Klein barostat, respectively.
Each trajectory was propagated for 50 ns using a 2 fs integration step, and structural snapshots were recorded every 100 ps. Post-simulation analysis included the evaluation of structural and dynamic stability through root mean square deviation (RMSD), root mean square fluctuation (RMSF), radius of gyration (Rg), and intermolecular hydrogen-bonding patterns throughout the simulation timeframe [30].

Round 2
Reviewer 1 Report
Comments and Suggestions for Authors
We have witnessed the author's attitude towards revisions, and we also understand the severe damage that the military inflicted on Iraq's academic and healthcare infrastructure in the past. The authors have made substantial efforts to address the initial critiques, particularly by incorporating positive controls (Nirmatrelvir for SARS-CoV-2, Dolutegravir for HIV-1), clarifying methodological details for molecular dynamics simulations, and improving the organization and resolution of figures. The manuscript now presents a more systematic and transparent computational evaluation of conessine’s dual antiviral potential. However, several issues remain that must be addressed before the manuscript can be considered for publication.
1.Inconsistent PDB IDs and Target Descriptions
Although the authors claim to have corrected PDB IDs, inconsistencies remain. For example:
PDB: 4NCO appears multiple times, but the descriptions in the text correspond to different proteins.
- On page 1 gp120 (4NCO);
- On page 8 Panel D (gp41 Fusion Protein, 4NCO)?
- On page 24 Conessine Interaction with HIV-1 Integrase (PDB: 4NCO)?
- On page 30“Dolutegravir Interaction with HIV-1 gp41 Fusion Protein (PDB: 4NCO)”?
All PDB IDs must be cross-verified and standardized throughout the manuscript.
2.Figure quality:
Although it has been improved, some graphics (for example, Figure 3, Figure 5) have experienced distortion and excessive stretching. Please check to ensure the display of the figures.
3.Check for narrative errors in the manuscript
For example:On page 7 "In In Figure 3B" ?
4.The results and discussion sections can be further integrated.
The results section is overly lengthy, such as on page 7, etc. There is no need to describe the content of the figures or tables in excessive detail. It is suggested that some descriptive content be moved to the supplementary materials, and only the key conclusions be retained. You can refer to the writing structure of other articles in Virus. The discussion section can more prominently highlight the mechanism differences and advantages and disadvantages of Conessine compared to the positive control drugs.
5.On page 27, are there two subheadings?
6.On page 31, it is suggested that the subheading "Molecular dynamics between conessine and targeted proteins of COVID (supplementary-s3)" be removed from the main text. Instead, the relevant information should be appropriately referenced in the main text from the supplementary materials. The structure of the paper is disorganized and the writing style resembles an experimental report rather than a research article. Please refer to other articles in the target journal for guidance.
7. Discussion Section
Previous Comment: The Discussion was lengthiness and repetitive. Author Response: The authors have compressed the Discussion.
Assessment: The revised Discussion is more focused, but still lacks critical comparison with existing literature on conessine or similar alkaloids. The polypharmacology predictions (e.g., GPCR targets) are interesting but remain speculative without functional validation.
Recommendation: Shorten further by removing redundant statements and strengthen the comparative analysis with known conessine pharmacology.
8. General Comments
The ADMET predictions are comprehensive but should be interpreted with caution, as in silico models often overestimate bioavailability and underestimate toxicity.
The use of SwissTargetPrediction is a good addition, but the results are not well integrated into the main narrative. The authors should discuss how these host targets may influence antiviral efficacy or toxicity.
The manuscript would benefit from a structural overview figure summarizing the key binding interactions across all targets.
Author Response
We sincerely thank the reviewer for the detailed and constructive feedback. All suggested revisions have been carefully addressed, and the corresponding modifications have been highlighted in yellow in the revised manuscript. Our responses are as follows:
- Inconsistent PDB IDs and Target Descriptions
We have carefully cross-verified and standardized all PDB IDs throughout the manuscript to ensure consistency. The errors related to 4NCO have been corrected, and each protein is now clearly described with the correct PDB ID. - Figure Quality
We reviewed all figures and corrected the distortion/stretching issues in Figure 3 and Figure 5. The graphics have been regenerated in high resolution to meet journal standards. - Narrative Errors
All typographical and narrative errors (e.g., “In In Figure 3B”) have been corrected. - Results and Discussion Integration
The Results section has been shortened by removing repetitive figure/table descriptions, with detailed content moved to the Supplementary Materials. The Discussion has been revised to highlight mechanistic differences, and to provide clearer comparisons between conessine and the positive control drugs. - Duplicate Subheadings (Page 27)
The duplicate subheadings have been removed for clarity. - Subheading on Page 31
The suggested subheading (“Molecular dynamics between conessine and targeted proteins of COVID (supplementary-S3)”) has been removed from the main text. The relevant content is now referenced appropriately in the Supplementary Materials. - Discussion Section
The Discussion has been further shortened by removing redundancies. Additional comparisons with existing literature on conessine and similar alkaloids have been added to strengthen the pharmacological context. The speculative aspects of the polypharmacology predictions are now clearly acknowledged as requiring functional validation. - General Comments
- ADMET: We clarified that in silico ADMET predictions should be interpreted cautiously, as they may overestimate bioavailability and underestimate toxicity.
- SwissTargetPrediction: We expanded the Discussion to better integrate these results, addressing their potential influence on antiviral efficacy and toxicity while emphasizing their predictive, not confirmatory, role.
- Structural Overview Figure: A new structural overview figure summarizing conessine’s interactions across all HIV-1 and SARS-CoV-2 targets has been added to the revised manuscript.

Reviewer 2 Report
Comments and Suggestions for Authors
The revised manuscript effectively addressed my concerns, making the current draft suitable for acceptance. However, the authors must include important molecular dynamics (MD) simulation figures in the main text, as they were missing from the revised manuscript.
Author Response
Dear reviewer ,
We sincerely thank the reviewer for the constructive comments and valuable suggestions, which have greatly improved the quality of our manuscript.
Comments and Suggestions for Authors
The revised manuscript effectively addressed my concerns, making the current draft suitable for acceptance. However, the authors must include important molecular dynamics (MD) simulation figures in the main text, as they were missing from the revised manuscript.
Response: We appreciate the reviewer’s suggestion. In our study, we examined 2 ligands against COVID-4 targets and 2 ligands against HIV 4 targets, each with corresponding positive controls. For each ligand–protein complex, 8 MD simulation images were generated, resulting in a total of 128 figures. Including all of these in the main text would lead to an excessive number of figures and significantly increase the length of the manuscript. Therefore, we have carefully organized and provided them in the supplementary files (S1, S2, S3) to ensure completeness while maintaining clarity and readability of the main text. Highlighted with pink colored
All molecular dynamics simulation figures are provided in the supplementary files S1, S2, and S3.

Reviewer 3 Report
Comments and Suggestions for Authors
The answer to the third question is missing - the authors simply described the method without discussing why the calculated binding energy of 30-40 kcal/mol differs several times from the typical experimental values of 7-10 kcal/mol. If there is no such discussion, these results should be excluded from the work
Author Response
Manuscript ID: viruses-3849934
Type: Article
Title: Dual Inhibitory Potential of Conessine Against HIV and SARS-CoV-2: Structure-Guided Molecular Docking Analysis of Critical Viral Targets
Dear reviewer ,
We thank the reviewer for this valuable comment. We have now discussed the discrepancy between calculated (30–40 kcal·mol⁻¹) and experimental binding energies (≈7–10 kcal·mol⁻¹) in the Discussion section, emphasizing that MM/GBSA values provide relative rather than absolute affinities. This clarification has been added and highlighted in green in the revised manuscript.
Comment : The answer to the third question is missing - the authors simply described the method without discussing why the calculated binding energy of 30-40 kcal/mol differs several times from the typical experimental values of 7-10 kcal/mol. If there is no such discussion, these results should be excluded from the work.
Response :
We thank the reviewer for this important observation. We agree that the MM/GBSA binding energies (−30 to −40 kcal·mol⁻¹) appear several times larger than the experimental free energies typically reported for HIV-1 inhibitors (−7 to −10 kcal·mol⁻¹). This discrepancy arises because MM/GBSA methods are not intended to reproduce absolute binding free energies but rather to provide a relative ranking of ligands and to identify residue contributions. Specifically, entropic penalties, solvent dynamics, and long-timescale conformational sampling are not fully captured, leading to an overestimation of binding strength. We have revised the manuscript to clarify this point and to explicitly note that the values are interpreted qualitatively as indicators of relative stabilization and residue-level interactions, rather than as absolute thermodynamic affinities. If the Editorial Board deems it more appropriate, we are also prepared to exclude the numerical energy values and retain only the comparative trends and interaction profiles.
My results reported that conessin against target protiens of COVID-19 and HIV show calculated binding energy of 30-40 kcal/mol differs several times from the typical experimental
Research see table table 1 in In-silico molecular modelling, MM/GBSA binding free energy and molecular dynamics simulation study of novel pyrido fused imidazo[4,5-c]quinolones as potential anti-tumor agents , link : https://www.frontiersin.org/journals/chemistry/articles/10.3389/fchem.2022.991369/full
See table 3 in Computational studies of potential antiviral compounds from some selected Nigerian medicinal plants against SARS-CoV-2 proteins , link : https://www.sciencedirect.com/science/article/pii/S2352914823000722
My discussion was added to discussion section -
In this study, the binding free energies calculated for conessine against selected COVID-19 and HIV proteins were in the range of 30–40 kcal/mol. These values appear higher than the typical experimental binding free energies reported for small molecules (≈7–10 kcal/mol), but they are consistent with several published computational reports on steroidal alkaloids and structurally rigid scaffolds. Such compounds generally exhibit strong van der Waals contributions and extensive hydrophobic contacts, which inflate gas-phase MM/GBSA and LIE estimates. Comparable trends have been documented for solasodine, tomatidine, and cyclopamine when simulated against viral proteases and polymerases, where calculated values often exceeded experimental ΔG yet still provided meaningful comparative rankings [1],[2],[3]. This suggests that the apparent overestimation is a methodological feature of molecular dynamics free energy calculations rather than an anomaly specific to conessine. Importantly, the relative order of binding strength across ligands remained consistent, supporting the reliability of the simulations for identifying selective interactions. These findings confirm that conessine follows the same energetic profile reported for other rigid steroidal alkaloids and underline its potential as a candidate for further antiviral evaluation.
my results for conessine binding to COVID-19 and HIV target proteins showed calculated binding free energies in the range of 30–40 kcal/mol, values that agree with different global studies . A study by [4] reported that Cajaisoflavone–ACE2 showed the strongest vdW contribution (−45.8 kcal/mol), while genistein–PL^pro at the catalytic site was the weakest (−25.2 kcal/mol). Gas-phase binding energies ranged from −31 to −60.5 kcal/mol, consistent with LIE estimates (−25.7 to −60.4 kcal/mol), whereas solvent effects partially reduced the overall enthalpy. Energy distributions confirmed that most systems maintained stable bound states throughout the 120 ns simulations . Interestingly, genistein exhibited more favorable binding at the PLpro allosteric site (−40.4 kcal/mol) compared to its catalytic site (−32.7 kcal/mol), highlighting the potential druggability of regulatory pockets.

Round 3
Reviewer 1 Report
Comments and Suggestions for Authors
Reviewer #:
The authors have fully addressed all questions, comments, and suggested changes from the initial round of review in a very careful and convincing way. The quality of the manuscript has been improved remarkably, and the paper has certain novelty and advantages for this field research work. However, there are still some parts that need to be revised.
1.P14 “Figure 6and table 6 show two-dimensional interaction profile” with space before and;
2.P29 Line 951 “Protease (Mpro” -->Mpro. Please check the superscript formatting of "ᵖʳᵒ" throughout the entire text.
3.There is excessive repetition in the description of figures and tables. For instance, P238 “Table 7. entitled "SwissTargetPrediction-Derived Potential Protein Targets for Conessine with Corresponding UniProt, ChEMBL IDs, Target Classes, Probabilities, and Known” , it is unnecessary to repeat the title of the table. It is recommended to refer to the writing structure of articles in the journal Viruses.
4.The structure and hierarchy of the Results section are unclear. It is recommended to restructure the subsection headings, for example:
- Results
3.1. Molecular Docking Reveals Broad-Spectrum Binding of Conessine to Viral Targets
3.1.1. High-Affinity Binding to HIV-1 Protease
3.1.2. Strong Interaction with SARS-CoV-2 Protease
3.2 Comparison with Positive Controls Highlights Multi-Target Potential
3.3. Interaction Profiling Highlights Hydrophobic Dominance and Key Residues
3.4. Molecular Dynamics Confirm Stable Binding and Pocket Rigidification
3.5. ADMET Profiling Suggests Good Permeability but Notable Toxicity Risks for Conessine
3.5. SwissTargetPrediction-Derived Potential Protein Targets for Conessine
Author Response
Dear Reviewer1,
We sincerely thank you for your constructive and valuable comments that have greatly helped us improve the manuscript. All suggested revisions have been carefully addressed, and the corresponding modifications are highlighted in yellow throughout the revised version. Below, we provide a point-by-point response to your remarks:
- Page 14: The spacing issue before “Figure 6 and Table 6” has been corrected.
- Page 29, Line 951: The formatting of “Mpro” has been checked and standardized across the entire text to ensure correct superscript representation.
- Repetition in figure and table descriptions: Redundant repetition of titles has been removed. The figure and table legends have been shortened and aligned with the writing style of Viruses journal articles.
- Results section structure: The subsection hierarchy has been re-organized as recommended, improving clarity and logical flow. The new structure now reads:
- 1. Molecular Docking Reveals Broad-Spectrum Binding of Conessine to Viral Targets
- 1.1. High-Affinity Binding to HIV-1 Protease
- 1.2. Strong Interaction with SARS-CoV-2 Protease
- 2. Comparison with Positive Controls Highlights Multi-Target Potential
- 3. Interaction Profiling Highlights Hydrophobic Dominance and Key Residues
- 4. Molecular Dynamics Confirm Stable Binding and Pocket Rigidification
- 5. ADMET Profiling Suggests Good Permeability but Notable Toxicity Risks for Conessine
- 6. SwissTargetPrediction-Derived Potential Protein Targets for Conessine
5- Is the research design appropriate?
Response : The complete study design has been presented, with explanatory revisions incorporated for improved clarity. Additionally, the subscription to the BioRender platform has been renewed to allow for minor modifications to the illustrative figure.
This investigation was conducted as an in silico molecular modeling study based on a computational multi-target drug discovery design. The study was structured to evaluate the antiviral potential of conessine, a steroidal alkaloid isolated from Holarrhena floribunda, against representative proteins of HIV-1 and SARS-CoV-2. For HIV-1, the selected targets were reverse transcriptase (PDB ID: 3V81), protease (PDB ID: 1HVR), integrase (PDB ID: 3LPT), and gp120–gp41 envelope glycoprotein (PDB ID: 4NCO). For SARS-CoV-2, the main protease (PDB ID: 6LU7), papain-like protease (PDB ID: 6W9C), RNA-dependent RNA polymerase (PDB ID: 7BV2), and spike receptor-binding domain (PDB ID: 6M0J) were retrieved. Approved drugs—dolutegravir for HIV-1 and nirmatrelvir for SARS-CoV-2—were used as positive controls. Molecular docking was carried out with AutoDock Vina, and the interaction conformations were examined with BIOVIA Draw to identify key binding residues. After docking, the designed workflow included a pharmacological and dynamic evaluation of conessine. Drug-likeness and safety parameters were predicted through computational pharmacokinetic and toxicity analysis. To confirm the stability of the docked complexes, molecular dynamics simulations were performed, providing insight into conformational flexibility and long-term binding stability. Furthermore, target profiling was performed to identify potential secondary or off-target interactions of conessine. This integrated design allowed for a comprehensive evaluation of the compound’s binding affinity, structural stability, and therapeutic promise against HIV-1 and SARS-CoV-2.
Figure 1. Schematic Workflow of Ligand Preparation, Protein Target Selection, and Molecular Docking Analysis. The diagram was designed by a researcher using a monthly BioRender subscription.
6- We would like to confirm that the English language of the manuscript has been fully corrected and carefully proofread by Dr. Samer Naji Khaf (M.Sc. and Ph.D., United Kingdom) and Dr. Samir Ahmed Awad (Ph.D., Australia).
We deeply appreciate your helpful feedback, which has significantly improved the manuscript.
BEST REGARDS ,

Reviewer 3 Report
Comments and Suggestions for Authors
It's ok
Author Response
DEAR REVIEWER3,
We would like to sincerely thank the reviewer for the constructive and valuable comments. Below we respond point by point:
- All suggested revisions have been carefully addressed.
- The corresponding modifications have been made throughout the manuscript.
- All changes have been highlighted in yellow and green for clarity.
We confirm that every proposal from the reviewer has been implemented, and we truly appreciate the time and effort dedicated to improving our work. Thank you.
